# Best-of-All-Worlds Bounds for Online Learning with Feedback Graphs

**Liad Erez**
Blavatnik School of Computer Science
Tel Aviv University
liaderez1@gmail.com

**Tomer Koren**
Blavatnik School of Computer Science
Tel Aviv University and Google Research
tkoren@tauex.tau.ac.il

## Abstract

We study the online learning with feedback graphs framework introduced by Mannor and Shamir [24], in which the feedback received by the online learner is specified by a graph $G$ over the available actions. We develop an algorithm that simultaneously achieves regret bounds of the form: $\mathcal{O}(\sqrt{\theta(G)T})$ with adversarial losses; $\mathcal{O}(\theta(G)\operatorname{polylog} T)$ with stochastic losses; and $\mathcal{O}(\theta(G)\operatorname{polylog} T + \sqrt{\theta(G)C})$ with stochastic losses subject to $C$ adversarial corruptions. Here, $\theta(G)$ is the *clique covering number* of the graph $G$. Our algorithm is an instantiation of Follow-the-Regularized-Leader with a novel regularization that can be seen as a product of a Tsallis entropy component (inspired by Zimmert and Seldin [27]) and a Shannon entropy component (analyzed in the corrupted stochastic case by Amir et al. [3]), thus subtly interpolating between the two forms of entropies. One of our key technical contributions is in establishing the convexity of this regularizer and controlling its inverse Hessian, despite its complex product structure.

## 1 Introduction

Online learning models a repeated interaction between a learner and an environment. In this framework, the learner has to choose an arm (or action) from a set of $N$ arms, iteratively across $T$ rounds. After choosing an arm the learner incurs an associated loss and receives feedback from the environment. The feedback the learner receives can range between the *full-information* setting where after choosing an action the learner sees the losses of all $N$ arms, and the *multi-armed bandit* (MAB) setting where only the loss of the arm chosen at the current round is revealed. Traditionally, the way the losses are generated is either in an adversarial manner (i.e., can be arbitrary in each round), or stochastically (i.e., sampled i.i.d. from an unknown distribution).

Recently, online algorithms achieving best-of-both-worlds guarantees have drawn significant attention: these are algorithms that are able to perform optimally in both the stochastic and adversarial settings, without prior knowledge of the regime and its parameters. Recent advances in this line of work [3, 27, 13, 28, 18] have resulted with algorithms that achieve a remarkable *best-of-all-worlds* guarantee: in a general intermediate adversarially-corrupted regime, where losses are generated stochastically but then may be modified by an adversary in an arbitrary way, these algorithms attain regret that grows with the square-root of the total amount of corruption introduced. Even more, these guarantees significantly outperform those obtained by specialized algorithms designed exclusively for the corrupted stochastic setting [22, 11, 23].

Our goal in this work is to extend these best-of-all-worlds results from the full-information and MAB settings to more general online learning problems and feedback models. We focus on the general framework of online learning with graph-structure feedback, originally introduced by Mannor and Shamir [24] and studied extensively since (e.g., [1, 10, 16, 9, 21, 20, 12]). In this model, feedback is described by an undirected graph $G$ over the arms; the set of observations received following each prediction is comprised of the loss of the chosen arm as well as the loss of all of its neighbors in $G$.

35th Conference on Neural Information Processing Systems (NeurIPS 2021).

The feedback graphs framework nicely interpolates between online learning with full-information and learning with bandit feedback, and generalizes to settings with more complex side-observation structure. Full-information online learning is captured as an extreme case where the graph $G$ is the clique over the $N$ arms; on the other extreme, an empty feedback graph that contains no edges corresponds to the multi-armed bandit problem.

The optimal regret in the feedback graphs setting is well understood, both in the adversarial and stochastic regimes. In the former, the state-of-the-art is obtained by Alon et al. [1] who gave an algorithm with regret $\mathcal{O}(\sqrt{\alpha(G)T})$, where $\alpha(G)$ is the independence number of $G$, and also established its optimality up to log factors. In the latter regime, Cohen et al. [10] proposed an algorithm whose regret is roughly $\mathcal{O}(\alpha(G)\log T)$, which is again nearly optimal. However, it remains unclear whether there exists a single algorithm that attains both bounds simultaneously, without prior knowledge of the regime, considering that the existing optimal algorithms in the two settings are inherently different. Of course, one can always ignore the additional feedback and use an existing best-of-all-worlds algorithm for MAB; however, this would result with a direct dependence on the number of arms $N$. The challenge is then to improve this dependence by exploiting the structure of the feedback graph.

## 1.1 Our contributions

We make a significant step towards obtaining a best-of-all-worlds guarantee in the feedback graphs model, by giving the first algorithm in this setting that achieves a uniform best-of-all-worlds guarantee and improves over the existing MAB bounds in a non-trivial way. Our algorithm achieves regret roughly of the form $\theta(G)\text{polylog}(T)$ when the losses are stochastic i.i.d., and $\widetilde{\mathcal{O}}(\sqrt{\theta(G)T})$ when the losses are fully adversarial; here $\theta(G)$ denotes the *clique covering number* of the graph $G$.[1] Furthermore, in an intermediate adversarially-corrupted stochastic setting, where the losses are generated stochastically but then may be corrupted by an adversary, we obtain a bound roughly of the form $\theta(G)\text{polylog}(T) + \widetilde{\mathcal{O}}(\sqrt{\theta(G)C})$ with $C$ being the total amount of corruption, that smoothly interpolates between these two extremes. Crucially, the algorithm achieves the aforementioned bounds simultaneously and agnostically, in the sense that it need not be aware of the actual underlying loss generation process, and in particular, of the actual level of adversarial corruption introduced.

To state our results more concretely, let $V_1, \ldots, V_\theta$ be a minimum clique cover of $G$ (here $\theta = \theta(G)$), and denote by $\Delta_k$ the minimal gap of a suboptimal arm in $V_k$; namely, the difference in mean losses (with respect to the stochastic uncorrupted losses) between an arm in $V_k$ and the best arm whose mean loss is minimal. Finally, define $Z = \sum_{k:\Delta_k>0} 1/\Delta_k$ (this is the quantity governing the complexity of the learning problem in the purely stochastic case; see [10]). Then, our main result can be stated as:

**Theorem** (main, informal). *There exists an algorithm (see Algorithm 1 in Section 4) which, given a clique covering $V_1, \ldots, V_\theta$ of $G$, obtains an expected regret bound of $\widetilde{\mathcal{O}}(\min\{\sqrt{\theta T}, Z + \sqrt{CZ}\})$ in the $C$-corrupted stochastic setting. In particular, in the stochastic setting ($C = 0$) the expected regret of the algorithm is at most $\widetilde{\mathcal{O}}(\sum_{k:\Delta_k>0} 1/\Delta_k)$, while in the adversarial setting ($C = T$) its expected regret is bounded by $\widetilde{\mathcal{O}}(\sqrt{\theta T})$.*

The above results leverage the graph structure in a non-trivial way and improve the direct dependence on the number of arms $N$ in existing best-of-all-worlds results to a dependence on $\theta(G)$, which is potentially significantly smaller than $N$. On the other hand, our bounds do not feature the optimal dependence on the independence number $\alpha(G)$; we leave it as a main open question whether it is possible to obtain a best-of-all-worlds result with $\alpha(G)$ replacing $\theta(G)$ in the bounds. We remark however, as we discuss in more detail below, that obtaining bounds scaling with the weaker $\theta(G)$ is already a highly non-trivial task and so we believe that our results form a significant step towards obtaining optimal best-of-all-worlds bounds. Another question we leave open is whether our dependence on $T$ can be improved to $\mathcal{O}(\log T)$ and $\mathcal{O}(\sqrt{T})$ in the stochastic and adversarial cases respectively, removing a few excess logarithmic factors in our bounds.

We note that the algorithm achieving the bound above is not adaptive to the best clique covering of the feedback graph for which the quantity $Z$ above is minimal. In other words, the algorithm requires a specific minimum clique covering as input, and the regret bounds we show take into account the suboptimality gaps with respect to this given covering, which may be worse than other coverings

---

[1]The clique covering number of a graph is the minimum number of cliques that cover its set of vertices; without loss of generality, we can assume that the latter set of cliques is a partition.

with respect to these gaps. Improving the algorithm such that it becomes adaptive to the minimum clique covering could be an interesting direction for future research.

## 1.2 Overview of main ideas and techniques

Our approach is inspired by recent progress in obtaining best-of-all-worlds guarantees in the two extreme cases of our problem: multi-armed bandits [27] and full-information experts [3]. Somewhat surprisingly, it has been shown in both cases that this type of guarantee can be obtained by simple algorithms based on the canonical Follow-the-Regularized-Leader template, traditionally used exclusively in adversarial online learning. Our primary technical challenge therefore boils down to designing a convex regularizer given the feedback graph $G$, that carefully interpolates between the well-understood regularization schemes in the full-information and bandit cases.

In the full-information case, best-of-all-worlds results [3] rely on (a time-varying version of) the negative Shannon entropy $R(p) = \sum_{i=1}^{N} p_i \log p_i$ as regularization. In a nutshell, its crucial properties are an $\mathcal{O}(\log N)$ uniform upper bound on its magnitude over the probability simplex, and on the other hand, that its Hessian $\nabla^2 R(p) = \text{diag}(p_1^{-1}, \ldots, p_N^{-1})$ satisfies $\ell^\top (\nabla^2 R(p))^{-1} \ell = \mathcal{O}(1)$ for any loss vector $\ell \in [0,1]^N$. (We omit more technical details on third-order conditions and self-bounding properties from this informal discussion.) In the bandit case, on the other hand, the best-of-all-worlds result of [27] relies on a Tsallis entropy (with parameter $\alpha = 1/2$) regularizer $R(p) = -\sum_{i=1}^{N} \sqrt{p_i}$, originally proposed in the context of MAB by [4]. This form of entropy is bounded uniformly over the simplex by $\mathcal{O}(\sqrt{N})$, and its Hessian $\nabla^2 R(p) = \frac{1}{4} \text{diag}(p_1^{-3/2}, \ldots, p_N^{-3/2})$ satisfies $\mathbb{E}[\tilde{\ell}^\top (\nabla^2 R(p))^{-1} \tilde{\ell}] = \mathcal{O}(\sqrt{N})$ for loss estimators $\tilde{\ell}$ used in MAB. In both cases, the regularizer strikes the "right" balance between the two bounds, which is necessary for optimal regret.

Moving to the more complex feedback model induced by a graph, it is natural to expect that an appropriate regularization would emerge as a certain combination of the Shannon and Tsallis entropies. For simplicity, consider a simple graph formed as a union of $\theta$ disjoint cliques $V_1, \ldots, V_\theta$, and let $p$ be a probability vector over its vertices. Then, we are looking for a regularization that would treat the marginal clique probabilities $q_k = \sum_{i \in V_k} p_i$ as a Tsallis entropy, while behaving within a clique (with respect to the internal conditional probabilities $p_i/q_k$ for $i \in V_k$) like the negative Shannon entropy. Inspecting the technical conditions more deeply, it turns out that we seek a regularizer $R$ such that the Hessian at $p$ is lower bounded by a diagonal matrix with the $i$'th diagonal entry corresponding to $i \in V_k$ is $p_i^{-1} q_k^{-1/2}$. A natural candidate would be simply the sum of the Shannon and Tsallis entropies; unfortunately, this regularization lacks the bi-level structure we need, and indeed, its Hessian does not admit the desired lower bound.

Our main technical innovation is the design of a new convex regularizer that admits the aforementioned lower bound on its Hessian, and on the other hand, is bounded by $\mathcal{O}(\sqrt{\theta} \log N)$ over the simplex. Interestingly, we find that the necessary conditions are met for a regularizer produced by the *product* of a negative Tsallis entropy (over the marginals $q_k$) and a negative Shannon entropy (over the conditional $p_i/q_k$), rather than their sum. Of course, such a product need not be a convex function in general, and indeed, the regularization we just described fails to be convex even in simple cases (see Fig. 1). Nevertheless, it turns out that there is a simple fix that makes this bi-level regularization a convex function: a simple linear shift to the Shannon entropy component. Not only that, but the resulting regularizer also admits the stronger Hessian lower bound we require. We refer to this new regularizer by *Tsallis-Shannon entropy*, and discuss the details of its derivation in Section 3; the remainder of the development takes more standard lines and is described in Sections 4 and 5.

## 1.3 Additional related work

The online learning with feedback graphs framework we consider here was introduced by Mannor and Shamir [24]. A tight characterization of the minimax regret in this model in the adversarial case was established by [2, 1]. The setting was first considered in the stochastic setting by Caron et al. [9], and was followed by numerous subsequent papers [10, 12, 20]. Cohen et al. [10] considered a setting in which the feedback can change in each time step, and the sequence of feedback graphs is not known to the algorithm, and Lu et al. [20] considered a setting where the loss distribution can change across rounds. Lee et al. [17] assume a hierarchical approach which maintains an explicit probability distribution over the cliques which is updated via OMD steps, whereas the individual arm probabilities within each clique are handled by a dedicated instance of adaptive Hedge. We remark that some of the mentioned work (most notably [1, 10]) analyze directed feedback graphs which are more general than the undirected graphs we consider here, and moreover, do not necessarily contain

self-loops. Such models currently lie beyond our scope, but we believe that our regularization techniques should be extensible to these variants too.

The line of work on best-of-both-worlds algorithms in the context of MAB was initiated by Bubeck and Slivkins [6]. Their result was subsequently improved in a sequence of papers [5, 25, 26], culminating in the remarkably elegant recent result of [27]. A number of tricks and techniques we use are borrowed (and sometimes extended) from this line of work, including refined regret bounds for FTRL with local norms [3, 27, 13], shifted loss estimators and self-bounding regret [26, 27, 3], and augmented log-barrier regularization for inducing stability [7, 17, 13]. The adversarially corrupted stochastic setting had also received considerable attention in recent years across numerous online learning frameworks [22, 23, 11, 14, 15, 19, 27, 13], including a paper by Lu et al. [21] who analyzed this setting in the bandits with graph feedback framework and showed a regret bound which generalizes naturally to the stochastic setting, but not to the adversarial setting. Thus the question of whether it is possible to obtain regret bounds which generalize to best-of-both-worlds type guarantees in the feedback graphs framework, was not resolved in these papers.

Finally, we note that the idea of using a hybrid regularization was suggested in a number of previous works. Bubeck et al. [7, 8] explored hybrid regularizers in the context of multi-armed bandits. More recent works include Zimmert et al. [28] who used a hybrid regularization closely related to Tsallis entropy presented in Zimmert and Seldin [27] to obtain best-of-both-worlds type guarantees in the semi-bandit setting. Similarly, Jin and Luo [13] use a form of a hybrid regularizer also related to Tsallis entropy in order to obtain similar guarantees for episodic reinforcement learning. However, to the best of our knowledge, our idea and analysis of a multi-level regularization that mixes together distinct types of entropies, were not previously explored.

## 2 Preliminaries

We consider an online learning problem where the learner has to repeatedly choose an arm from a set of $N$ arms (or actions) indexed by $[N] = \{1, 2, ..., N\}$. In each time step $t = 1, 2, ..., T$ the learner generates a probability vector $p_t$ from the $N$-dimensional simplex $\mathcal{S}_N = \{p \in \mathbb{R}_+^N : \sum_{i=1}^N p_i = 1\}$ and then chooses an arm $I_t \sim p_t$. Thereafter, a loss vector $\ell_t \in [0,1]^N$ is generated, the learner incurs the loss $\ell_{t,I_t}$, and feedback is revealed.

**Feedback model:** The feedback received by the learner at each round $t$ is specified by an undirected *feedback graph* $G = ([N], E)$, known to the learner in advance, and is comprised of $\{(i, \ell_{t,i}) : i \in \mathcal{N}(I_t)\}$, where $\mathcal{N}(j)$ denotes the set of neighbors of node $j$ in $G$ (which we assume always to contain $j$ itself). Moreover, we assume that the learner receives as input a minimum *clique covering* of $G$, i.e., a minimum cardinality collection of cliques $\{V_1, V_2, ..., V_\theta\}$ which partitions and spans the vertices of $G$, meaning each $i \in [N]$ belongs to some unique clique $V_k \subseteq [N]$. The number of cliques $\theta$ in this minimum clique covering, also denoted by $\theta(G)$, is called the *clique covering number* of $G$.[2]

**Assumption on losses:** We consider a general *adversarially-corrupted stochastic* setting (originally introduced in [22]) that includes stochastic and adversarial online learning as special cases, as well as a range of intermediate problems. In this setting, loss vectors $\tilde{\ell}_1, \tilde{\ell}_2, ..., \tilde{\ell}_T$ are first drawn i.i.d. from a fixed probability distribution, unknown to the learner. We denote the mean loss vector by $\mu = (\mu_1, \mu_2, ..., \mu_N) = \mathbb{E}[\tilde{\ell}_t]$, and let $i^\star = \arg\min_i \mu_i$ the best arm, which we assume to be unique. We further let $V_{k^\star} = V(i^\star)$ and denote by $\delta_i = \mu_i - \mu_{i^\star}$ the gap between arm $i$ and the best arm, and $\Delta_k = \min_{i \in V_k, i \neq i^\star} \delta_i$ the minimal gap of a suboptimal arm which belongs to $V_k$ (here, if $V(i^\star)$ is a singleton then we interpret $\Delta_{k^\star} = 0$). After the stochastic loss vectors $\tilde{\ell}_1, \ldots, \tilde{\ell}_T$ have been generated, an adversary is allowed to corrupt them and form a final loss sequence $\ell_1, \ldots, \ell_T \in [0,1]^N$. The overall (expected) *corruption level* introduced by the adversary is defined as

$$C = \mathbb{E}\left[\sum_{t=1}^T \|\ell_t - \tilde{\ell}_t\|_\infty\right].$$

Importantly, we assume that this quantity is unknown to the learner, and the algorithms we design will be oblivious to the value of $C$.

---

[2]Computing a minimum clique covering in general graphs is a well-known NP-hard problem; we therefore assume it is given as an input, to avoid dealing with such computational efficiency issues.

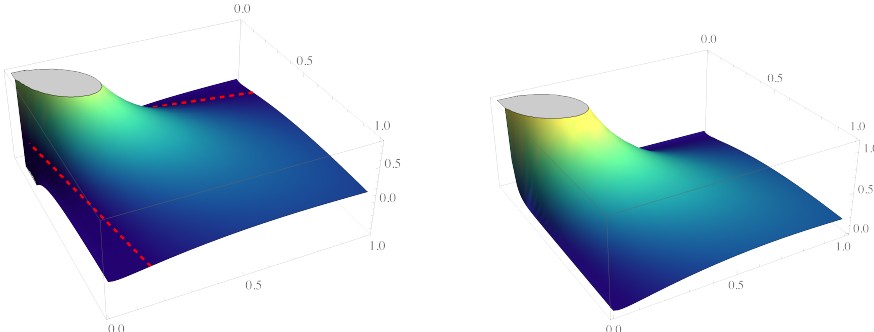

Figure 1: The minimal eigenvalue of the Hessian of the Tsallis-perspective in $d = 2$ dimensions (see Eq. (1)) of $h = \psi_\alpha$ (see Eq. (2)) with $\alpha = 0$ (left) vs. $\alpha = 0.25$ (right). The dashed red lines indicate the zero level set. We see that convexity of $\psi_\alpha$ is insufficient to ensure that $\Psi_\alpha$ is even convex (let alone strongly convex), and the linear shift introduced in $\psi_\alpha$ is crucial for this purpose.

**Regret:** The goal of the learner in the adversarially-corrupted stochastic setting is to minimize the *pseudo-regret*, defined as follows:

$$\mathcal{R}_T \triangleq \mathbb{E}\left[\sum_{t=1}^T \ell_{t,I_t}\right] - \min_{i \in [N]} \mathbb{E}\left[\sum_{t=1}^T \ell_{t,i}\right] = \mathbb{E}\left[\sum_{t=1}^T p_t \cdot \ell_t\right] - \min_{i \in [N]} \mathbb{E}\left[\sum_{t=1}^T \ell_{t,i}\right].$$

Here, the expectation is over the internal randomness of the algorithm and over any randomness in the losses. The *stochastic* setting is obtained for corruption level $C = 0$, where the pseudo-regret simplifies to $\mathcal{R}_T = \sum_{t=1}^T \sum_{i=1}^N \mathbb{E}[p_{t,i}]\delta_i$. The *adversarial* setting is recovered by setting $C = T$; with an oblivious adversary (that sets the loss vectors ahead of time), the loss vectors are w.l.o.g. deterministic and the above definition of pseudo-regret matches the usual definition of regret.

**Additional notation:** Given $i \in [N]$ and a partition $V_1, \ldots, V_\theta$ of $[N]$, we denote by $V(i)$ the (unique) partition element which $i$ belongs to, i.e., $i \in V(i)$. For a vector $p \in \mathbb{R}^N$ and a set of arms $A \subseteq [N]$, we use the shorthand $p(A) = \sum_{i \in A} p_i$.

## 3 The Tsallis-Shannon Entropy

In this section we introduce our definition of the Tsallis-Shannon entropy that we use as a regularizer and derive the main properties we require. For that, we first define the notion of the Tsallis-perspective of a convex function, that will be useful for this derivation.

### 3.1 Tsallis-perspective of a convex function

Let $h : [0, 1] \to \mathbb{R}$ be twice-differentiable and strictly convex. We define the *Tsallis-perspective* of $h$ as the following function over $\mathbb{R}_+^d$:

$$H(x) \triangleq \sqrt{\|x\|_1} \sum_{i=1}^d h\left(\frac{x_i}{\|x\|_1}\right). \tag{1}$$

The name means to draw a connection to the classical notion of the perspective of a convex function $h$, given by $g(x, t) = th(x/t)$, which is always convex in the pair $(x, t)$; using this fact, the convexity of the function $G(x) = \|x\|_1 \sum_{i=1}^d h(x_i/\|x\|_1)$ immediately follows since $x \mapsto \|x\|_1$ is linear over $\mathbb{R}_+^d$. In contrast, for the similarly looking function $H$, that has the leading $\|x\|_1$ factor under a square-root, an analogous result does not hold in general; see Fig. 1 for an example. Remarkably, however, there is a simple condition on $h$ under which it can be shown that not only $H$ is convex, but in fact admits a strong lower bound over its Hessian. This is detailed in the following lemma, whose proof is in the full version of the paper [**?** ].

**Lemma 1.** *Assume that $h : [0, 1] \to \mathbb{R}$ is twice-differentiable, strictly convex and satisfies the following condition for some constants $c_h \in \mathbb{R}$ and $\lambda_h > 0$, for all vectors $y \in Y \subseteq \mathcal{S}_d$:*

$$\sum_{i=1}^d h(y_i) + 2 \sum_{i=1}^d \frac{(h'(y_i) - c_h)^2}{h''(y_i)} + \lambda_h \leq 0.$$

*Then, at any $x \in \mathbb{R}_+^d$ such that $x/\|x\|_1 \in Y$, the function $H$ satisfies*

$$\nabla^2 H(x) \succeq \frac{\lambda_h}{4} \|x\|_1^{-\frac{3}{2}} J + \frac{1}{2} \|x\|_1^{-\frac{7}{2}} \sum_{i=1}^{d} x_i^2 h'' \left( \frac{x_i}{\|x\|_1} \right) z_i z_i^{\mathsf{T}},$$

*where $J$ is the $d$-by-$d$ all-ones matrix and $z_i = \mathbf{1}_d - (\|x\|_1/x_i)\mathbf{e}_i$ for $i \in [d]$.*

### 3.2 Deriving the Tsallis-Shannon entropy

We can now define the Tsallis-Shannon entropy in terms of Tsallis-perspectives and derive its local strong convexity properties that we require for our analysis. For a given $\alpha > 0$, the Tsallis-Shannon entropy with respect to a partition $V_1, \ldots, V_\theta$ of $V = [N]$ is defined for all $p \in \mathbb{R}_+^d$ as

$$\Psi_\alpha(p) = \sum_{k=1}^{\theta} \sqrt{p(V_k)} \cdot \sum_{i \in V_k} \psi_\alpha \left( \frac{p_i}{p(V_k)} \right), \quad \text{where} \quad \psi_\alpha(y) = y \log y - \alpha y . \tag{2}$$

Namely, $\Psi_\alpha$ is the sum of the Tsallis-perspectives of $\psi_\alpha$ with respect to each partition element $V_k$. Observe that $\psi_\alpha$ corresponds to a single term of a (linearly shifted) negative Shannon entropy. Thus, for a probability vector $p$, the function $\Psi_\alpha$ can be viewed as a bi-level entropy, where the top level amounts to the marginal probabilities $p(V_k)$ (for $k \in [\theta]$) and the bottom level to the conditional probabilities $p_i/p(V_k)$ (for $i \in V_k$); for the marginal probabilities $\Psi_\alpha$ behaves like a $\frac{1}{2}$-Tsallis entropy, while on the conditional probabilities it operates as a (negative, shifted) Shannon entropy.

It is not hard to show that the magnitude of $\Psi_\alpha(p)$ is at most $\mathcal{O}(\sqrt{\theta} \log N)$, which is crucial for our purposes as it avoids a direct polynomial dependence on the dimension $N$. On the other hand, using Lemma 1, we can prove the following lower bound for the Hessian of $\Psi_\alpha$ for an appropriate choice of $\alpha$. We remark that the setting of $\alpha$ is crucial: the linear shift in the Shannon entropy component is essential even just for ensuring the convexity of $\Psi_\alpha$ (see Fig. 1).

**Lemma 2.** *Fix $\gamma > 0$, and let $p \in \mathbb{R}_+^N$ such that $p_i/p(V_k) \geq \gamma$ for all $i, k$ with $i \in V_k$. Then for $\alpha = 2(1 + \log^2(1/\gamma))$, the Hessian of $\Psi_\alpha$ at $p$ satisfies*

$$\nabla^2 \Psi_\alpha(p) \succeq \frac{1}{2} \operatorname{diag} \left( \frac{1}{p_1 \sqrt{p(V(1))}}, \frac{1}{p_2 \sqrt{p(V(2))}}, \ldots, \frac{1}{p_N \sqrt{p(V(N))}} \right).$$

*Proof.* For brevity, we omit all $\alpha$ subscripts below. Observe that $\Psi$ is a separable sum of $\theta$ functions, each of which is a function of variables $p_i$ with $i \in V_k$, thus its Hessian $\nabla^2 \Psi(p)$ is block-diagonal with blocks aligned with $V_1, \ldots, V_\theta$. It therefore suffices to establish the Hessian bound for a function of the form $\Psi^\circ(x) = \sqrt{\|x\|_1} \sum_{i=1}^{d} \psi(x_i/\|x\|_1)$ corresponding to a set of variables $V = \{x_1, \ldots, x_d\}$. To this end, we first show that the function $\psi$ satisfies the conditions of Lemma 1 over the domain $Y = \mathcal{S}_d \cap \{y : y_i \geq \gamma, \ \forall i \in [d]\}$, with constants $c_\psi = -(1 + 2\log^2(1/\gamma))$ and $\lambda_\psi = 2$. Indeed, twice differentiability and strict convexity are immediate, and for any $y \in Y$ we have

$$\sum_{i=1}^{d} \psi(y_i) + 2 \sum_{i=1}^{d} \frac{(\psi'(y_i) - c_\psi)^2}{\psi''(y_i)} + \lambda_\psi \leq -2\log^2 \tfrac{1}{\gamma} + 2 \sum_{i=1}^{d} y_i \left( \log y_i - 1 - 2\log^2 \tfrac{1}{\gamma} - c_\psi \right)^2$$

$$= -2\log^2 \tfrac{1}{\gamma} + 2 \sum_{i=1}^{d} y_i \log^2 y_i$$

$$\leq 0.$$

Applying the latter lemma on $\Psi^\circ$, we obtain (below $J$ denotes the $d$-by-$d$ all-ones matrix):

$$\nabla^2 \Psi^\circ(x) \succeq \frac{1}{2} \|x\|_1^{-\frac{3}{2}} J + \frac{1}{2} \|x\|_1^{-\frac{7}{2}} \sum_{i=1}^{d} x_i^2 \psi'' \left( \frac{x_i}{\|x\|_1} \right) z_i z_i^{\mathsf{T}}$$

$$= \frac{1}{2} \|x\|_1^{-\frac{3}{2}} J + \frac{1}{2} \|x\|_1^{-\frac{5}{2}} \sum_{i=1}^{d} x_i z_i z_i^{\mathsf{T}}$$

$$= \frac{1}{2}\|x\|_1^{-\frac{3}{2}}J + \frac{1}{2}\|x\|_1^{-\frac{3}{2}}J - \frac{1}{2}\|x\|_1^{-\frac{3}{2}}\sum_{i=1}^{d}\left(\mathbf{1}_d\mathbf{e}_i^\mathsf{T} + \mathbf{e}_i\mathbf{1}_d^\mathsf{T}\right) + \frac{1}{2}\|x\|_1^{-\frac{1}{2}}\sum_{i=1}^{d}\frac{1}{x_i}\mathbf{e}_i\mathbf{e}_i^\mathsf{T}$$

$$= \|x\|_1^{-\frac{3}{2}}J - \frac{1}{2}\|x\|_1^{-\frac{3}{2}}(2J) + \frac{1}{2}\|x\|_1^{-\frac{1}{2}}\operatorname{diag}\left(\frac{1}{x_1}, \frac{1}{x_2}, ..., \frac{1}{x_d}\right)$$

$$= \frac{1}{2\sqrt{\|x\|_1}}\operatorname{diag}\left(\frac{1}{x_1}, \frac{1}{x_2}, ..., \frac{1}{x_d}\right),$$

and the result follows. ∎

## 4 Algorithm and Main Result

In this section we present our best-of-all-worlds algorithm for online learning with feedback graphs. The algorithm, detailed in Algorithm 1, follows the general Follow-the-Regularized-Leader (FTRL) template, and is instantiated by a choice of time-varying convex regularization functions $R_t$ together with a standard loss estimator for graph-structured feedback due to [2].

---

**Algorithm 1:** FTRL with feedback graphs

**Input:** clique covering $\{V_1, V_2, ..., V_\theta\}$ of an undirected feedback graph $G$;
let $\alpha = 2(\log^2(NT) + 1)$, $\beta = 9$, $\gamma = 1/(NT)$, and step sizes $\eta_t = 1/\sqrt{t}$;
initialize $\widehat{L}_{0,i} = 0$ for all $i \in [N]$;
**for** $t = 1, 2, ..., T$ **do**
    update
$$p_t = \arg\min_{p \in \mathcal{S}_N^\gamma}\left\{\widehat{L}_{t-1} \cdot p + R_t(p)\right\}$$
    where $R_t(p) = \eta_t^{-1}\Psi(p) + \Phi(p)$ ($\Psi$ and $\Phi$ are defined in Eqs. (3) and (4));
    pick arm $I_t \sim p_t$, observe feedback $\{(i, \ell_{t,i}) : i \in \mathcal{N}(I_t)\}$ and set
$$\forall i \in [N]: \qquad \widehat{\ell}_{t,i} = \frac{\ell_{t,i}}{p_t(V(i))}\mathbb{I}\{i \in V(I_t)\};$$
    update $\widehat{L}_t = \widehat{L}_{t-1} + \widehat{\ell}_t$;
**end**

---

Our main result regarding Algorithm 1 is given in the following.

**Theorem 1.** *Algorithm 1 attains the following expected pseudo-regret bound in the C-corrupted stochastic setting:*

$$\mathcal{R}_T = \widetilde{\mathfrak{O}}\left(\min\left\{\sqrt{\theta T}, \sum_{k:\Delta_k > 0}\frac{1}{\Delta_k} + \sqrt{C\sum_{k:\Delta_k > 0}\frac{1}{\Delta_k}}\right\}\right).$$

The regularizer $R_t$ is formed as a sum of two convex functions, $\Psi$ and $\Phi$, described below. Predictions are computed by the algorithm by minimizing the cumulative estimated loss, regularized by $R_t$, over a truncated simplex $\mathcal{S}_N^\gamma \triangleq \{p \in \mathcal{S}_N : \forall i, \ p_i \geq \gamma\}$. For loss estimation, the algorithm relies on the standard unbiased loss estimators (denoted by $\widehat{\ell}_t$) for the graph-feedback framework. Note however that while the feedback revealed to the algorithm at time step $t$ includes the losses of all the neighbors of $I_t$ in $G$, it only makes use of the losses of the arms which belong to the same clique as $I_t$; in other words, the algorithm may ignore some of the feedback it receives.

The primary regularization function $\Psi$ (whose weight increases with time) is the Tsallis-Shannon entropy $\Psi_\alpha$ defined in Section 3, given explicitly as

$$\Psi(p) = -\alpha\sum_{k=1}^{\theta}\sqrt{p(V_k)} + \sum_{k=1}^{\theta}\frac{1}{\sqrt{p(V_k)}}\sum_{i \in V_k}p_i\log\frac{p_i}{p(V_k)}. \tag{3}$$

The second time-invariant regularizer $\Phi$ is a log-barrier of the marginal clique probabilities,

$$\Phi(p) = -\beta \sum_{k=1}^{\theta} \log p(V_k), \tag{4}$$

and its goal is to further stabilize the algorithm. While the idea of augmenting the regularizer with a log-barrier function for promoting stability is not new, we note that a standard log-barrier of the form $-\sum_{i=1}^{N} \log p_i$ would inevitably introduce an additive $\mathcal{O}(N)$ to the regret, which is suboptimal for our purposes. For that reason we use a different variant of a log-barrier function that operates on the marginal probabilities of the cliques.

## 5 Proof of Theorem 1

In this section we provide details on the proof of our main theorem. Our main step towards proving Theorem 1 is the following general regret bound for Algorithm 1.

**Theorem 2.** *Algorithm 1 attains the following pseudo-regret bound, regardless of the corruption level:*

$$\mathcal{R}_T = \widetilde{\mathcal{O}}\left(\theta + \sum_{t=1}^{T} \sum_{k \neq k^\star} \sqrt{\frac{\mathbb{E}[p_t(V_k)]}{t}} + \sum_{t=1}^{T} \sqrt{\frac{\mathbb{E}[p_t(V_{k^\star} \setminus i^\star)]}{t}} + \sum_{t=1}^{T} \sqrt{\frac{\mathbb{E}[p_t^+(V_{k^\star} \setminus i^\star)]}{t}}\right), \tag{5}$$

*where $p_t^+ = \arg\min_{p \in \mathcal{S}_N^\gamma} \{\widehat{L}_t \cdot p + R_t(p)\}$ for all $t$.*

The focus of this section is on proving Theorem 2, but first we sketch how it implies our main result.

*Proof of Theorem 1 (sketch; full proof in the full version of the paper [? ])* The worst-case bound of $\widetilde{\mathcal{O}}(\sqrt{\theta T})$ follows immediately from the fact that $\sum_{t=1}^{T} \sum_{k=1}^{\theta} \sqrt{\mathbb{E}[p_t(V_k)]/t} \leq 2\sqrt{\theta T}$ via Jensen's inequality, and similarly for the term including $p_t^+$. The bound involving the corruption level $C$ requires a more delicate argument due to [27] that makes use of a self-bounding property of the pseudo-regret. In more detail, using Young's inequality one has for all $z > 0$:

$$\sum_{t=1}^{T} \sum_{k \neq k^\star} \sqrt{\frac{\mathbb{E}[p_t(V_k)]}{t}} \leq \sum_{t=1}^{T} \sum_{k \neq k^\star} \left(\frac{z}{2t\Delta_k} + \frac{\mathbb{E}[p_t(V_k)]\Delta_k}{2z}\right),$$

and a similar bound can be shown for the other two summations in Eq. (5). Combining these two gives after some simplification the pseudo-regret bound $\mathcal{R}_T \leq zB + z^{-1}(\mathcal{R}_T + 2C)$, for $B = \mathcal{O}(\sum_{k:\Delta_k > 0} \log(T)/\Delta_k)$, which further simplifies to $\mathcal{R}_T \leq 2B + (z-1)B + \frac{2C+B}{z-1}$. Optimizing the bound with respect to $z$ then gives $\mathcal{R}_T \leq 4B + 4\sqrt{BC}$, which implies the bound we claimed. ∎

We now set to prove Theorem 2. Our starting point is a general regret bound for FTRL. Applying a variation on the standard FTRL analysis (a similar analysis was used in [27, 13]) to the FTRL instance of Algorithm 1 gives the following regret bound.

**Lemma 3.** *For all $p^\gamma \in \mathcal{S}_N^\gamma$ the following holds:*

$$\sum_{t=1}^{T} \widehat{\ell}_t \cdot (p_t - p^\gamma) \leq \Phi(p^\gamma) - \Phi(p_1) + \sum_{t=1}^{T} \left(\frac{1}{\eta_t} - \frac{1}{\eta_{t-1}}\right)(\Psi(p^\gamma) - \Psi(p_t)) \tag{6}$$

$$+ 2 \sum_{t=1}^{T} \eta_t \left(\|\widehat{\ell}_t - \ell_{t,i^\star} \mathbf{1}\|_t^*\right)^2. \tag{7}$$

*Here $\|g\|_t^* = \sqrt{g^\top (\nabla^2 \Psi(\tilde{p}_t))^{-1} g}$ is the dual local norm induced by $\Psi$ at $\tilde{p}_t$ for some intermediate point $\tilde{p}_t \in [p_t, p_t^+]$, where $p_t^+ = \arg\min_{p \in \mathcal{S}_N^\gamma} \{\widehat{L}_t \cdot p + R_t(p)\}$.*

For the proof, see the full version of the paper [? ]. The bound above can be seen as a sum of a *penalty* term and a *stability* term (RHS of Eqs. (6) and (7) respectively). In order to prove Theorem 2 using this general regret bound, we separately bound the penalty and stability terms. The penalty term is

simpler to handle, and we only provide a brief discussion about how it is bounded. Bounding the stability term requires more work and constitutes the bulk of our analysis and our main contributions. We remark that the stability term contains dual norms of the loss estimators after introducing an additive shift of the form $\ell_{t,i^\star}$ to each arm. The purpose of such a shift is to exclude the contribution of the best arm $i^\star$ from the regret bound, as can be seen in Eq. (5). More details regarding this shift and its purpose are given in the full version of the paper [**?**].

**Bounding the penalty term:** We present a brief discussion about bounding the *penalty* term in Eq. (6). The penalty term is handled using bounds on the magnitude of the regularization over the domain $\mathcal{S}_N^\gamma$.

**Lemma 4.** *The penalty term (RHS of Eq. (6)) is upper bounded by*

$$\widetilde{\mathfrak{O}}\left(\theta + \sum_{t=1}^{T}\sum_{k\neq k^\star}\sqrt{\frac{p_t(V_k)}{t}} + \sum_{t=1}^{T}\sqrt{\frac{p_t(V_{k^\star}\setminus i^\star)}{t}}\right),$$

*for $p^\gamma$ defined by $p_i^\gamma = \begin{cases} \gamma & i \neq i^\star \\ 1-(N-1)\gamma & i = i^\star \end{cases}$.*

*Proof (sketch; full proof in the full version of the paper [**?**])* Essentially, we use the fact that the log-barrier component of the regularization is bounded by $\widetilde{\mathfrak{O}}(\theta)$, and the Tsallis-Shannon component evaluated on the prediction $p_t$ is bounded by $\widetilde{\mathfrak{O}}(\sum_{k=1}^{\theta}\sqrt{p_t(V_k)})$, as it can be seen as a weighted sum of entropy functions (bounded in magnitude by $\log N$), each of which corresponds to a clique $V_k$ with the respective weight being $\sqrt{p_t(V_k)}$. The non-trivial part in bounding the penalty term is in omitting the term corresponding to $V_{k^\star}$ from the sum. This is accomplished by our specific choice of $p^\gamma$ described above, as well as carefully bounding the (Shannon) entropy terms within each clique. ∎

**Bounding the stability term:** The following lemma provides an upper bound on the stability term (RHS of Eq. (7)). This lemma crucially relies on strong convexity properties of the Tsallis-Shannon regularization (Eq. (3)). These properties allow us to obtain highly non-trivial upper bounds on the eigenvalues of the inverse Hessian of $\Psi(\cdot)$ over $\mathcal{S}_N^\gamma$, and our ability to obtain such bounds turns out to be critical for us to be able to successfully bound the stability term.

**Lemma 5.** *The following holds for all time steps $t$:*

$$\mathbb{E}\left[(\|\widehat{\ell}_t - \ell_{t,i^\star}\mathbf{1}\|_t^*)^2\right] = \widetilde{\mathfrak{O}}\left(\sum_{k\neq k^\star}\sqrt{\mathbb{E}\left[p_t(V_k)\right]} + \sqrt{\mathbb{E}\left[p_t^+(V_{k^\star}\setminus i^\star)\right]}\right).$$

*Proof (sketch; full proof in the full version of the paper [**?**])* For simplicity, in this proof sketch we ignore the shift in the loss estimation and only bound $\mathbb{E}[(\|\widehat{\ell}_t\|_t^*)^2]$ as it captures the main challenges and new ideas in the analysis. First, recall that the dual norm is formally defined with respect to the Hessian of $\Psi(\cdot)$ at some intermediate point $\tilde{p}_t \in [p_t, p_t^+]$. We use the stability properties induced by the augmented log-barrier regularizer to replace $\tilde{p}_t(V_i)$ terms with $p_t(V_i)$ up to constant factors, and within each clique, bound the remaining terms involving $\tilde{p}_i$ by a sum of similar terms with the latter replaced by $p_i$ and $p_i^+$.

Next, for bounding the relevant dual norm, we require a suitable upper bound on $(\nabla^2\Psi(p_t))^{-1}$. To this end, we employ Lemma 2, by which $\nabla^2\Psi(p_t)$ is lower bounded by a diagonal matrix $D_t$, in which the $i$'th diagonal entry corresponding to $i \in V_k$ is of the form $(2\sqrt{p_t(V_k)}p_{t,i})^{-1}$. We now use this crucial fact to obtain the following upper bound on the stability term:

$$\mathbb{E}\left[(\|\widehat{\ell}_t\|_t^*)^2\right] \leq 2\mathbb{E}\left[\widehat{\ell}_t^\top D_t^{-1}\widehat{\ell}_t\right] = 2\mathbb{E}\left[\sum_{k=1}^{\theta}\sqrt{p_t(V_k)}\sum_{i\in V_k}p_{t,i}\widehat{\ell}_{t,i}^2\right].$$

We keep bounding this term by using the definition of the loss estimators $\widehat{\ell}_t$ to obtain

$$\mathbb{E}\left[(\|\widehat{\ell}_t\|_t^*)^2\right] \leq 2\mathbb{E}\left[\sqrt{p_t(V(I_t))}\sum_{i\in V(I_t)}p_{t,i}\left(\frac{\ell_{t,i}}{p_t(V(I_t))}\right)^2\right] \leq 2\mathbb{E}\left[\frac{1}{\sqrt{p_t(V(I_t))}}\right] = 2\mathbb{E}\left[\sum_{k=1}^{\theta}\sqrt{p_t(V_k)}\right],$$

where in the last line we used the fact that the probability that at time step $t$ the algorithm chooses an arm from the clique $V_k$ is $p_t(V_k)$. Note that this bound contains a term for $p_{t,i^\star}$ which we would like to omit from the final bound. As we remarked before, the way we mitigate that is by considering the shifted loss estimators. This is explained in full in the full version of the paper [**?** ]. ∎

**Concluding the proof:** We can now sketch the proof of Theorem 2 using the bounds we obtained. A formal proof that also takes into account the poly-log factors can be found in the full version of the paper [**?** ].

*Proof of Theorem 2 (sketch; full proof in the full version of the paper [**?** ])* We first remark that it suffices to bound the expected regret with respect to $p^\gamma$ given by $\mathbb{E}\big[\sum_{t=1}^{T} \ell_t \cdot (p_t - p^\gamma)\big]$, since it can only be larger than the pseudo-regret by an additive constant, as shown in the full version of the paper [**?** ]. In addition, since the loss estimators $\widehat{\ell}_t$ are unbiased estimators of the loss vectors $\ell_t$ we conclude that it suffices to bound the regret with respect to the loss estimators, i.e. $\mathbb{E}\big[\sum_{t=1}^{T} \widehat{\ell}_t \cdot (p_t - p^\gamma)\big]$ which is bounded by the sum of the expected penalty and stability terms (Eqs. (6) and (7)). Lemma 4 gives a bound on the penalty term of the form

$$\widetilde{\mathfrak{O}}\!\left(\theta + \sum_{t=1}^{T}\sum_{k \neq k^\star} \sqrt{\frac{\mathbb{E}[p_t(V_k)]}{t}} + \sum_{t=1}^{T}\sqrt{\frac{\mathbb{E}[p_t(V_{k^\star}\setminus i^\star)]}{t}}\right),$$

and Lemma 5 gives a bound on the stability term of the form

$$\widetilde{\mathfrak{O}}\!\left(\sum_{t=1}^{T}\sum_{k \neq k^\star} \sqrt{\frac{\mathbb{E}[p_t(V_k)]}{t}} + \sum_{t=1}^{T}\sqrt{\frac{p_t^+(V_{k^\star}\setminus i^\star)}{t}}\right).$$

Adding the two bounds and rearranging, we conclude the proof of Theorem 2. ∎

### Acknowledgements and Funding Disclosure

This work has received support from the Israeli Science Foundation (ISF) grant no. 2549/19, from the Len Blavatnik and the Blavatnik Family foundation, and from the Yandex Initiative in Machine Learning.

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
