# Supplementary Material

## A Tsallis-perspective: Proofs

To prove Lemma 1 we need the following technical result that gives an expression for the Hessian of the Tsallis-perspective $H$ in terms of the (scalar) derivatives of $h$.

**Lemma 6.** *The Hessian of $H$ (Eq. (1)) at any point $x \in \mathbb{R}_+^d$ can be expressed as:*

$$
\nabla^2 H(x) = -\frac{1}{4} \|x\|_1^{-\frac{3}{2}} \sum_{i=1}^d h\left(\frac{x_i}{\|x\|_1}\right) z z^\mathsf{T}
$$
$$
+ \|x\|_1^{-\frac{7}{2}} \sum_{i=1}^d x_i^2 h''\left(\frac{x_i}{\|x\|_1}\right) z_i z_i^\mathsf{T}
$$
$$
+ \frac{1}{2} \|x\|_1^{-\frac{5}{2}} \sum_{i=1}^d x_i h'\left(\frac{x_i}{\|x\|_1}\right)\left(z z_i^\mathsf{T} + z_i z^\mathsf{T}\right),
$$

*where $z = \mathbf{1}_d$ is the all-ones vector, and $z_i = \mathbf{1}_d - (\|x\|_1/x_i)\mathbf{e}_i$ for all $i \in [d]$.*

*Proof.* Let us first compute the first and second derivatives of $f(x) = \sqrt{\|x\|_1}$ and $g_i(x) = h(x_i/\|x\|_1)$ for a fixed $i \in [d]$:

$$
\nabla f(x) = \frac{1}{2} \|x\|_1^{-\frac{1}{2}} z;
$$
$$
\nabla^2 f(x) = -\frac{1}{4} \|x\|_1^{-\frac{3}{2}} z z^\mathsf{T};
$$
$$
\nabla g_i(x) = h'\left(\frac{x_i}{\|x\|_1}\right)\left(\frac{1}{\|x\|_1}\mathbf{e}_i - \frac{x_i}{\|x\|_1^2} z\right)
$$
$$
= -\frac{x_i}{\|x\|_1^2} h'\left(\frac{x_i}{\|x\|_1}\right) z_i;
$$
$$
\nabla^2 g_i(x) = h''\left(\frac{x_i}{\|x\|_1}\right)\left(\frac{1}{\|x\|_1}\mathbf{e}_i - \frac{x_i}{\|x\|_1^2} z\right)\left(\frac{1}{\|x\|_1}\mathbf{e}_i - \frac{x_i}{\|x\|_1^2} z\right)^\mathsf{T}
$$
$$
+ h'\left(\frac{x_i}{\|x\|_1}\right)\left(-\frac{1}{\|x\|_1^2} z \mathbf{e}_i^\mathsf{T} - \frac{1}{\|x\|_1^2}\mathbf{e}_i z^\mathsf{T} + \frac{2 x_i}{\|x\|_1^3} z z^\mathsf{T}\right)
$$
$$
= \frac{x_i^2}{\|x\|_1^4} h''\left(\frac{x_i}{\|x\|_1}\right) z_i z_i^\mathsf{T} + \frac{x_i}{\|x\|_1^3} h'\left(\frac{x_i}{\|x\|_1}\right)\left(z z_i^\mathsf{T} + z_i z^\mathsf{T}\right).
$$

Using the formula for the Hessian of a product, we now obtain:

$$
\nabla^2\left(f(x) g_i(x)\right)
$$
$$
= \left(\nabla^2 f(x)\right) g_i(x) + \nabla f(x)\nabla g_i(x)^\mathsf{T} + \nabla g_i(x)\nabla f(x)^\mathsf{T} + f(x)\left(\nabla^2 g_i(x)\right)
$$
$$
= -\frac{1}{4} \|x\|_1^{-\frac{3}{2}} h\left(\frac{x_i}{\|x\|_1}\right) z z^\mathsf{T} - \frac{1}{2} \|x\|_1^{-\frac{5}{2}} x_i h'\left(\frac{x_i}{\|x\|_1}\right)\left(z z_i^\mathsf{T} + z_i z^\mathsf{T}\right)
$$
$$
+ \|x\|_1^{-\frac{7}{2}} x_i^2 h''\left(\frac{x_i}{\|x\|_1}\right) z_i z_i^\mathsf{T} + \|x\|_1^{-\frac{5}{2}} x_i h'\left(\frac{x_i}{\|x\|_1}\right)\left(z z_i^\mathsf{T} + z_i z^\mathsf{T}\right)
$$
$$
= -\frac{1}{4} \|x\|_1^{-\frac{3}{2}} h\left(\frac{x_i}{\|x\|_1}\right) z z^\mathsf{T} + \|x\|_1^{-\frac{7}{2}} x_i^2 h''\left(\frac{x_i}{\|x\|_1}\right) z_i z_i^\mathsf{T} + \frac{1}{2} \|x\|_1^{-\frac{5}{2}} x_i h'\left(\frac{x_i}{\|x\|_1}\right)\left(z z_i^\mathsf{T} + z_i z^\mathsf{T}\right).
$$

Summing this over $i = 1, \ldots, d$, we obtain the expression for the Hessian $\nabla^2 H(x)$. ∎

*Proof of Lemma 1.* Fix $x \in \mathbb{R}_+^d$ and let $y_i = x_i / \|x\|_1$ for all $i$. By Lemma 6 the Hessian of $H$ can be written as

$$\nabla^2 H(x) = \frac{1}{4} \|x\|_1^{-\frac{3}{2}} \sum_{i=1}^d \left( -h(y_i) z z^\mathsf{T} + 4 y_i^2 h''(y_i) z_i z_i^\mathsf{T} + 2 y_i h'(y_i) \left( z z_i^\mathsf{T} + z_i z^\mathsf{T} \right) \right),$$

where $z = \mathbf{1}_d$ is the all-ones vector, and $z_i = \mathbf{1}_d - (\|x\|_1 / x_i) \mathbf{e}_i$ for all $i \in [d]$. Then, using the condition on $h$ and since $\sum_{i=1}^d y_i z_i = 0$ we have

$$
\begin{aligned}
\nabla^2 H(x) &\geq \frac{\lambda_h}{4} \|x\|_1^{-\frac{3}{2}} J + \frac{1}{4} \|x\|_1^{-\frac{3}{2}} \sum_{i=1}^d \left( \frac{(h'(y_i) - c_h)^2}{\frac{1}{2} h''(y_i)} z z^\mathsf{T} + 4 y_i^2 h''(y_i) z_i z_i^\mathsf{T} + 2 y_i h'(y_i) \left( z z_i^\mathsf{T} + z_i z^\mathsf{T} \right) \right) \\
&= \frac{\lambda_h}{4} \|x\|_1^{-\frac{3}{2}} J + \frac{1}{4} \|x\|_1^{-\frac{3}{2}} \sum_{i=1}^d \left( \frac{(h'(y_i) - c_h)^2}{\frac{1}{2} h''(y_i)} z z^\mathsf{T} + 4 y_i^2 h''(y_i) z_i z_i^\mathsf{T} + 2 y_i (h'(y_i) - c_h) \left( z z_i^\mathsf{T} + z_i z^\mathsf{T} \right) \right) \\
&= \frac{\lambda_h}{4} \|x\|_1^{-\frac{3}{2}} J + \frac{1}{4} \|x\|_1^{-\frac{3}{2}} \sum_{i=1}^d \left( \frac{h'(y_i) - c_h}{\sqrt{\frac{1}{2} h''(y_i)}} z + 2 y_i \sqrt{\tfrac{1}{2} h''(y_i)} z_i \right) \left( \frac{h'(y_i) - c_h}{\sqrt{\frac{1}{2} h''(y_i)}} z + 2 y_i \sqrt{\tfrac{1}{2} h''(y_i)} z_i \right)^\mathsf{T} \\
&\quad + \frac{\lambda_h}{4} \|x\|_1^{-\frac{3}{2}} \sum_{i=1}^d y_i^2 h''(y_i) z_i z_i^\mathsf{T},
\end{aligned}
$$

and the result follows since each term in the first summation is psd. $\blacksquare$

## B  Proof of Main Result

In this section we provide the proof of Theorem 1. In Appendix B.1 we prove useful lemmas which provide us with stability properties of the FTRL iterates. In Appendix B.2 and Appendix B.3 we bound the stability and penalty terms (RHS of Eq. (6) and Eq. (5)) towards proving Theorem 2 in Appendix B.4. We then prove Theorem 1 in Appendix B.5.

### B.1  Stability of Iterates

We first establish a technical stability property of the FTRL updates that is crucial for bounding the stability term (Eq. (6)). This property asserts that for every time step $t$, the clique marginal probabilities induced by $p_t$ are close, up to a constant multiplicative factor, to the clique marginals induced by $p_t^+$, where $p_t^+ \triangleq \arg\min_{p \in \mathcal{S}_N^\gamma} \{ \widehat{L}_t \cdot p + R_t(p) \}$. The proof uses properties of the log-barrier component $\Phi$, and relies on an adaptation of an argument of Jin and Luo [13].

**Lemma 7.** *For all time steps $t$ and cliques $V_k$ it holds that $p_t^+(V_k) \leq \frac{7}{3} p_t(V_k)$, where $p_t^+ \triangleq \arg\min_{p \in \mathcal{S}_N^\gamma} \{ \widehat{L}_t \cdot p + R_t(p) \}$.*

*Proof.* We define:

$$
\begin{aligned}
F_t(p) &= \widehat{L}_{t-1} \cdot p + R_t(p), \\
F_t^+(p) &= \widehat{L}_t \cdot p + R_t(p),
\end{aligned}
$$

so that $p_t = \arg\min_{p \in \mathcal{S}_N^\gamma} \{ F_t(p) \}$ and $p_t^+ = \arg\min_{p \in \mathcal{S}_N^\gamma} \{ F_t^+(p) \}$. Note that $\nabla^2 \Phi(p)$ is a block diagonal matrix, with the block corresponding to the clique $V_k$ being exactly $\frac{9}{p(V_k)^2} J_{V_k}$ where $J_{V_k}$ is the $|V_k| \times |V_k|$ all-ones matrix. A straightforward calculation then shows that for all $p, p', p'' \in \mathcal{S}_N$ it holds that:

$$\|p' - p''\|_{\nabla^2 \Phi(p)}^2 = 9 \sum_{k=1}^K \frac{(p'(V_k) - p''(V_k))^2}{p(V_k)^2}.$$

It suffices to prove that $\|p_t^+ - p_t\|_{\nabla^2 \Phi(p_t)}^2 \leq 16$. This is because by the calculation we just made, we have $\left( p_t^+(V_k) - p_t(V_k) \right)^2 \leq \left( \frac{4}{3} p_t(V_k) \right)^2$ which is want we want to prove. It then suffices to show that for any $p' \in \mathcal{S}_N^\gamma$ with $\|p' - p_t\|_{\nabla^2 \Phi(p_t)}^2 = 16$ we have $F_t^+(p') \geq F_t^+(p_t)$. This is because as

an implication of that, $p_t^+$ which minimizes the convex function $F_t^+$, must be within the convex set $\{p : \|p - p_t\|^2_{\nabla^2\Phi(p_t)} \leq 16\}$. We proceed to lower bound $F_t^+(p')$ as follows:

$$
\begin{aligned}
F_t^+(p') &= F_t^+(p_t) + \nabla F_t^+(p_t)^{\mathsf{T}}(p' - p_t) + \frac{1}{2}\|p' - p_t\|^2_{\nabla^2 R_t(\xi)} \\
&= F_t^+(p_t) + \nabla F_t(p_t)^{\mathsf{T}}(p' - p_t) + \widehat{\ell_t^{\mathsf{T}}}(p' - p_t) + \frac{1}{2}\|p' - p_t\|^2_{\nabla^2 R_t(\xi)} \\
&\geq F_t^+(p_t) + \widehat{\ell_t^{\mathsf{T}}}(p' - p_t) + \frac{1}{2}\|p' - p_t\|^2_{\nabla^2\Phi(\xi)},
\end{aligned}
$$

where the first equality is a Taylor expansion of $F_t^+$ around $p_t$, with $\xi$ being a point between $p'$ and $p_t$, and the last inequality is due to first-order optimality conditions and the fact that $\nabla^2 R_t(\xi) \succeq \nabla^2\Phi(\xi)$ since $\Psi$ is convex. Note that since $\|p' - p_t\|^2_{\nabla^2\Phi(p_t)} = 16$, by the same argument as in the beginning of the proof we conclude that $p'(V_k) \leq \frac{7}{3}p_t(V_k)$. Since $\xi$ lies between $p_t$ and $p'$ we conclude the same ratio bound for $\xi$. We can thus bound the last term as follows:

$$
\begin{aligned}
\frac{1}{2}\|p' - p_t\|^2_{\nabla^2\Phi(\xi)} &= \frac{9}{2}\sum_{k=1}^{K} \frac{(p'(V_k) - p_t(V_k))^2}{(\xi(V_k))^2} \\
&\geq \frac{9}{2 \cdot \left(\frac{7}{3}\right)^2}\sum_{k=1}^{K} \frac{(p'(V_k) - p_t(V_k))^2}{p_t(V_k)^2} \\
&= \frac{9}{2 \cdot 49}\|p' - p_t\|^2_{\nabla^2\Phi(p_t)} \\
&= \frac{72}{49} \geq 1.
\end{aligned}
$$

It now suffices to show that $\widehat{\ell_t^{\mathsf{T}}}(p' - p_t) \geq -1$; indeed,

$$
\widehat{\ell_t^{\mathsf{T}}}(p' - p_t) = \sum_{i \in V(I_t)} \frac{\ell_{t,i}}{p_t(V(I_t))}(p'_{t,i} - p_{t,i}) \geq -\frac{1}{p_t(V(I_t))}\sum_{i \in V(I_t)} \ell_{t,i}p_{t,i} \geq -1,
$$

and the proof is complete. ∎

The following lemma showcases another stability property that relates $p_t$ to $p_t^+$. A corollary of this lemma is that the pseudo-regret of the iterates $p_t$ can only be larger than the pseudo-regret of the iterates $p_t^+$, and it is used in the proof of Theorem 1 in Appendix B.5.

**Lemma 8.** *For all time steps t it holds that*

$$
p_t^+ \cdot \widehat{\ell}_t \leq p_t \cdot \widehat{\ell}_t,
$$

*where $p_t^+ \triangleq \arg\min_{p \in \mathcal{S}_N^\gamma}\{\widehat{L}_t \cdot p + R_t(p)\}$.*

*Proof.* Since $p_t^+$ is a minimizer of $\widehat{L}_t \cdot p + R_t(p)$ and $p_t$ is a minimizer of $\widehat{L}_{t-1} \cdot p + R_t(p)$, we have:

$$
\begin{aligned}
\widehat{L}_t \cdot p_t^+ + R_t(p_t^+) &\leq \widehat{L}_t \cdot p_t + R_t(p_t) \\
&= \widehat{\ell}_t \cdot p_t + \widehat{L}_{t-1} \cdot p_t + R_t(p_t) \\
&\leq \widehat{\ell}_t \cdot p_t + \widehat{L}_{t-1} \cdot p_t^+ + R_t(p_t^+),
\end{aligned}
$$

and the claim follows by rearranging terms. ∎

## B.2 Proof of Lemma 5 (Stability)

We now restate Lemma 5 which bounds the stability term to include extra constants which appear in the bound.

**Lemma 5** (restated). *The following holds for all time steps t:*

$$\mathbb{E}\big[(\|\widehat{\ell}_t - \ell_{t,i^\star}\mathbf{1}\|_t^*)^2\big] = 56 \sum_{k \neq k^\star} \sqrt{\mathbb{E}\big[p_t(V_k)\big]} + 8\sqrt{\mathbb{E}\big[p_t^+(V_{k^\star} \setminus i^\star)\big]}.$$

*Here* $\|g\|_t^* = \sqrt{g^\top (\nabla^2 \Psi(\tilde{p}_t))^{-1} g}$ *is the dual local norm induced by* $\Psi$ *at* $\tilde{p}_t$ *for some intermediate point* $\tilde{p}_t \in [p_t, p_t^+]$, *where* $p_t^+ = \arg\min_{p \in \mathcal{S}_N^\gamma} \{\widehat{L}_t \cdot p + R_t(p)\}$.

*Proof.* By Lemma 2, $\nabla^2 \Psi(\tilde{p}_t)$ is lower bounded by a diagonal matrix $D_t$ in which the $i$'th diagonal entry corresponding to $i \in V_k$ is $\left(2\sqrt{\tilde{p}_t(V_k)}\tilde{p}_{t,i}\right)^{-1}$. Equivalently it holds that $\left(\nabla^2\Psi(\tilde{p}_t)\right)^{-1} \preceq D_t^{-1}$. Using this fact and the fact that $\widehat{\ell}_{t,i} = 0$ for $i \notin V(I_t)$ we have

$$\mathbb{E}\left[(\|\widehat{\ell}_t - \ell_{t,i^\star}\mathbf{1}\|_t^*)^2\right] = \mathbb{E}\left[(\widehat{\ell}_t - \ell_{t,i^\star}\mathbf{1})^\top\left(\nabla^2\Psi(\tilde{p}_t)\right)^{-1}(\widehat{\ell}_t - \ell_{t,i^\star}\mathbf{1})\right]$$

$$\leq 2\mathbb{E}\left[\sum_{k=1}^K \sqrt{\tilde{p}_t(V_k)} \sum_{i \in V_k} \tilde{p}_{t,i}(\widehat{\ell}_{t,i} - \ell_{t,i^\star})^2\right]$$

$$= 2\mathbb{E}\left[\sqrt{\tilde{p}_t(V(I_t))} \sum_{i \in V(I_t)} \tilde{p}_{t,i}(\widehat{\ell}_{t,i} - \ell_{t,i^\star})^2\right] \tag{7}$$

$$+ 2\mathbb{E}\left[\sum_{V_k \neq V(I_t)} \sqrt{\tilde{p}_t(V_k)} \sum_{i \in V_k} \tilde{p}_{t,i}(\ell_{t,i^\star})^2\right], \tag{8}$$

where in the final equality we split the sum over cliques into a term for $V(I_t)$ and a sum over the rest of the cliques. We first show that the RHS of Eq. (7) is bounded as follows:

$$\mathbb{E}\left[\sqrt{\tilde{p}_t(V(I_t))} \sum_{i \in V(I_t)} \tilde{p}_{t,i}(\widehat{\ell}_{t,i} - \ell_{t,i^\star})^2\right] \leq 16 \sum_{k \neq k^\star} \sqrt{\mathbb{E}[p(V_k)]} + 4\sqrt{\mathbb{E}\big[p_t^+(V_{k^\star} \setminus i^\star)\big]}.$$

Indeed, due to Lemma 7 and the fact that $\tilde{p}_t$ lies between $p_t$ and $p_t^+$ it holds that $\tilde{p}_t(V_k) \leq 3p_t(V_k)$ for all $k$. Plugging in the expression for the loss estimator $\widehat{\ell}_t$ we obtain

$$\mathbb{E}\left[\sqrt{\tilde{p}_t(V(I_t))} \sum_{i \in V(I_t)} \tilde{p}_{t,i}(\widehat{\ell}_{t,i} - \ell_{t,i^\star})^2\right] = \mathbb{E}\left[\sqrt{\tilde{p}_t(V(I_t))} \sum_{i \in V(I_t)} \tilde{p}_{t,i}\left(\frac{\ell_{t,i}}{p_t(V(I_t))} - \ell_{t,i^\star}\right)^2\right]$$

$$\leq 2\mathbb{E}\left[p_t(V(I_t))^{-\frac{3}{2}} \sum_{i \in V(I_t)} \tilde{p}_{t,i}(\ell_{t,i} - p_t(V(I_t))\ell_{t,i^\star})^2\right]$$

$$= 2\mathbb{E}\left[\sum_{k=1}^K p_t(V_k)^{-\frac{1}{2}} \sum_{i \in V_k} \tilde{p}_{t,i}(\ell_{t,i} - p_t(V_k)\ell_{t,i^\star})^2\right],$$

where in the last equality we use the law of total expectation and the fact that conditioned on the history up until time step $t$ (including the decision vector $p_t$), the probability that $I_t$ belongs to the clique $V_k$ is exactly $p_t(V_k)$. In more detail:

$$\mathbb{E}\left[p_t(V(I_t))^{-\frac{3}{2}} \sum_{i \in V(I_t)} \tilde{p}_{t,i}(\ell_{t,i} - p_t(V(I_t))\ell_{t,i^\star})^2\right]$$

$$= \mathbb{E}\left[\mathbb{E}_t\left[p_t(V(I_t))^{-\frac{3}{2}} \sum_{i \in V(I_t)} \tilde{p}_{t,i}(\ell_{t,i} - p_t(V(I_t))\ell_{t,i^\star})^2\right]\right]$$

$$= \mathbb{E}\left[\sum_{k=1}^K \Pr[I_t \in V_k \mid h_t] \cdot \mathbb{E}_t\left[p_t(V_k)^{-\frac{3}{2}} \sum_{i \in V_k} \tilde{p}_{t,i}(\ell_{t,i} - p_t(V_k)\ell_{t,i^\star})^2\right]\right]$$

$$= \mathbb{E}\left[\sum_{k=1}^{K} p_t(V_k) \cdot \mathbb{E}_t\left[p(V_k)^{-\frac{3}{2}} \sum_{i \in V_k} \tilde{p}_{t,i}\big(\ell_{t,i} - p_t(V_k)\ell_{t,i\star}\big)^2\right]\right]$$

$$= \mathbb{E}\left[\mathbb{E}_t\left[\sum_{k=1}^{K} p_t(V_k)^{-\frac{1}{2}} \sum_{i \in V_k} \tilde{p}_{t,i}\big(\ell_{t,i} - p_t(V_k)\ell_{t,i\star}\big)^2\right]\right]$$

$$= \mathbb{E}\left[\sum_{k=1}^{K} p_t(V_k)^{-\frac{1}{2}} \sum_{i \in V_k} \tilde{p}_{t,i}\big(\ell_{t,i} - p_t(V_k)\ell_{t,i\star}\big)^2\right],$$

where $h_t$ denotes the history up to and including the choice of $p_t$ at time step $t$ (not including the choice of $I_t$), and in the fourth equality we use linearity of expectation and the fact that $p_t(V_k)$ is constant when conditioned on $h_t$. We proceed to bound the above term, while splitting the sum over cliques into a term for $V_{k\star}$ and a sum for all of the other cliques:

$$\mathbb{E}\left[\sum_{k=1}^{K} p_t(V_k)^{-\frac{1}{2}} \sum_{i \in V_k} \tilde{p}_{t,i}\big(\ell_{t,i} - p_t(V_k)\ell_{t,i\star}\big)^2\right]$$

$$\leq \mathbb{E}\left[\sum_{k \neq k\star} p_t(V_k)^{-\frac{1}{2}} \tilde{p}_t(V_k)\right] + \mathbb{E}\left[p_t(V_{k\star})^{-\frac{1}{2}}\left(\sum_{i \in V_{k\star}, i \neq i\star} \tilde{p}_{t,i} + \tilde{p}_{t,i\star}(1 - p_t(V_{k\star}))^2\right)\right]$$

$$\leq 3\mathbb{E}\left[\sum_{k \neq k\star} \sqrt{p_t(V_k)}\right] + 2\mathbb{E}\left[\tilde{p}_t(V_{k\star})^{-\frac{1}{2}} \tilde{p}_t\big(V_{k\star} \setminus i\star\big)\right] + 3\mathbb{E}\left[(1 - p_t(V_{k\star}))^2\right]$$

$$\leq 6\mathbb{E}\left[\sum_{k \neq k\star} \sqrt{p_t(V_k)}\right] + 2\mathbb{E}\left[\sqrt{\tilde{p}_t(V_{k\star} \setminus i\star)}\right]$$

$$\leq 8\mathbb{E}\left[\sum_{k \neq k\star} \sqrt{p_t(V_k)}\right] + 2\mathbb{E}\left[\sqrt{p_t^+(V_{k\star} \setminus i\star)}\right]$$

$$\leq 8 \sum_{k \neq k\star} \sqrt{\mathbb{E}[p_t(V_k)]} + 2\sqrt{\mathbb{E}\big[p_t^+(V_{k\star} \setminus i\star)\big]},$$

where in the last inequality we used Jensen's inequality. We now proceed to bound the RHS of Eq. (8):

$$\mathbb{E}\left[\sum_{V_k \neq V(I_t)} \sqrt{\tilde{p}_t(V_k)} \sum_{i \in V_k} \tilde{p}_{t,i}(\ell_{t,i\star})^2\right] \leq \mathbb{E}\left[\sum_{V_k \neq V(I_t)} \tilde{p}_t(V_k)^{\frac{3}{2}}\right]$$

$$\leq 6\mathbb{E}\left[\sum_{V_k \neq V(I_t)} p_t(V_k)^{\frac{3}{2}}\right] \tag{9}$$

$$= 6\mathbb{E}\left[\sum_{k=1}^{K} (1 - p_t(V_k)) p_t(V_k)^{\frac{3}{2}}\right] \tag{10}$$

$$\leq 12\mathbb{E}\left[\sum_{k \neq k\star} \sqrt{p_t(V_k)}\right]$$

$$\leq 12 \sum_{k \neq k\star} \sqrt{\mathbb{E}[p_t(V_k)]},$$

where in Eq. (9) we use Lemma 7 and the fact that $\tilde{p}_t$ lies between $p_t$ and $p_t^+$, in Eq. (10) we use the fact that the probability of the clique $V_k$ not to be chosen at time step $t$ is $1 - p_t(V_k)$ and the last line uses Jensen's inequality. Combining the two bounds, we conclude the proof. ∎

### B.3 Proof of Lemma 4 (Penalty)

In this section we restate Lemma 4 which bounds the penalty term to include the extra constants and poly-log factors.

 **Lemma 4** (restated). *The penalty term described in the RHS of Eq. (5) is bounded by*

$$9K \log \frac{1}{\gamma} + 5 \log^2 \frac{1}{\gamma} \sum_{t=1}^{T} \sum_{k \neq k^\star} \sqrt{\frac{p_t(V_k)}{t}} + 2 \log \frac{1}{\gamma} \sum_{t=1}^{T} \sqrt{\frac{p_t(V_{k^\star} \setminus i^\star)}{t}}, \tag{11}$$

*where* $p_i^\gamma = \begin{cases} \gamma & i \neq i^\star \\ 1 - (N-1)\gamma & i = i^\star \end{cases}$ *for all* $i \in [N]$ *and* $\frac{1}{\eta_0} \triangleq 0$.

*Proof.* Noting that $\Phi(\cdot) \geq 0$ we can bound the first term as follows:

$$\Phi(p^\gamma) - \Phi(p_1) \leq \Phi(p^\gamma) \leq 9K \log \frac{1}{\gamma}.$$

Continuing with the second part of the penalty term, note that by definition of $p^\gamma$ we have $p^\gamma(V_{k^\star}) \geq p_t(V_{k^\star})$ for all $t$. Also note that $\Psi(p^\gamma) \leq -2\left(\log^2 \frac{1}{\gamma} + 1\right)\sqrt{p^\gamma(V_{k^\star})}$. We then have

$$\sum_{t=1}^{T} \left(\frac{1}{\eta_t} - \frac{1}{\eta_{t-1}}\right)(\Psi(p^\gamma) - \Psi(p_t)) \leq \sum_{t=1}^{T} \left(\sqrt{t} - \sqrt{t-1}\right)\left(2\left(\log^2 \frac{1}{\gamma} + 1\right)\sum_{k=1}^{K} \sqrt{p_t(V_k)}\right.$$

$$+ \sum_{k=1}^{K} \frac{1}{\sqrt{p_t(V_k)}} \sum_{i \in V_k} p_{t,i} \log \frac{p_t(V_k)}{p_{t,i}} - 2\left(\log^2 \frac{1}{\gamma} + 1\right)\sqrt{p^\gamma(V_{k^\star})}\right)$$

$$\leq 2\left(\log^2 \frac{1}{\gamma} + 1\right)\sum_{t=1}^{T} \frac{1}{\sqrt{t}} \sum_{k \neq k^\star} \sqrt{p_t(V_k)}$$

$$+ \log \frac{1}{\gamma} \sum_{t=1}^{T} \frac{1}{\sqrt{t}} \sum_{k \neq k^\star} \sqrt{p_t(V_k)}$$

$$+ \sum_{t=1}^{T} \frac{1}{\sqrt{t \cdot p_t(V_{k^\star})}} \sum_{i \in V_{k^\star}} p_{t,i} \log \frac{p_t(V_{k^\star})}{p_{t,i}}$$

$$\leq 5 \log^2 \frac{1}{\gamma} \sum_{t=1}^{T} \frac{1}{\sqrt{t}} \sum_{k \neq k^\star} \sqrt{p_t(V_k)}$$

$$+ \sum_{t=1}^{T} \frac{1}{\sqrt{t \cdot p_t(V_{k^\star})}} \sum_{i \in V_{k^\star}} p_{t,i} \log \frac{p_t(V_{k^\star})}{p_{t,i}}, \tag{12}$$

where the second inequality follows from the fact that $\frac{1}{\sqrt{t}} - \frac{1}{\sqrt{t-1}} \leq \frac{1}{\sqrt{t}}$ and that $p_{t,i} \geq \gamma$ for all $t$ and $i$. It is left to bound the final term. Using the inequality $\log x \leq x - 1$ for all $x > 0$ we have

$$\sum_{i \in V_{k^\star}} p_{t,i} \log \frac{p_t(V_{k^\star})}{p_{t,i}} = \sum_{i \in V_{k^\star} \setminus i^\star} p_{t,i} \log \frac{p_t(V_{k^\star})}{p_{t,i}} + p_{t,i^\star} \log \frac{p_t(V_{k^\star})}{p_{t,i^\star}}$$

$$\leq \log \frac{1}{\gamma} \sum_{i \in V_{k^\star} \setminus i^\star} p_{t,i} + p_{t,i^\star}\left(\frac{p_t(V_{k^\star})}{p_{t,i^\star}} - 1\right)$$

$$= \left(\log \frac{1}{\gamma} + 1\right)p_t\left(V_{k^\star} \setminus i^\star\right)$$

$$\leq 2 \log \frac{1}{\gamma} p_t\left(V_{k^\star} \setminus i^\star\right).$$

Plugging this bound into Eq. (12) while using the fact that $\frac{p_t(V_{k^\star} \setminus i^\star)}{\sqrt{p_t(V_{k^\star})}} \leq \sqrt{p_t(V_{k^\star} \setminus i^\star)}$ completes the proof. ∎

 **B.4   Proof of Theorem 2**

520  In order to prove Theorem 2 we make use of the following simple claim which asserts that the pseudo-
521  regret is bounded up to an additive constant factor by the regret with respect to some probability
522  vector in $\mathcal{S}_N^\gamma$.

523  **Lemma 9.** *For all $\gamma \in [0, \frac{1}{N}]$ and $i^\star \in [N]$ the following holds:*

$$\mathbb{E}\left[\sum_{t=1}^{T} p_t \cdot \widehat{\ell}_t - \mathbf{e}_{i^\star} \cdot \sum_{t=1}^{T} \widehat{\ell}_t\right] \le \mathbb{E}\left[\sum_{t=1}^{T} p_t \cdot \widehat{\ell}_t - p^\gamma \cdot \sum_{t=1}^{T} \widehat{\ell}_t\right] + \gamma TN,$$

524  *where* $p_i^\gamma = \begin{cases} \gamma & i \ne i^\star \\ 1 - (N-1)\gamma & i = i^\star \end{cases} \quad \forall i \in [N].$

525  *Proof.* Fix $\gamma \in [0, \frac{1}{N}]$ and $i^\star \in [N]$. Note that $\mathbf{e}_{i^\star} = p^\gamma - v$ where $v$ is defined as follows:

$$v_i = \begin{cases} \gamma & i \ne i^\star \\ -(N-1)\gamma & i = i^\star \end{cases} \quad \forall i \in [N].$$

526  This observation gives us the following:

$$\mathbb{E}\left[\sum_{t=1}^{T} p_t \cdot \widehat{\ell}_t - \mathbf{e}_{i^\star} \cdot \sum_{t=1}^{T} \widehat{\ell}_t\right] = \mathbb{E}\left[\sum_{t=1}^{T} p_t \cdot \ell_t - \mathbf{e}_{i^\star} \cdot \sum_{t=1}^{T} \ell_t\right]$$

$$= \mathbb{E}\left[\sum_{t=1}^{T} p_t \cdot \ell_t - p^\gamma \cdot \sum_{t=1}^{T} \ell_t\right] + v \cdot \mathbb{E}\left[\sum_{t=1}^{T} \ell_t\right]$$

$$= \mathbb{E}\left[\sum_{t=1}^{T} p_t \cdot \widehat{\ell}_t - p^\gamma \cdot \sum_{t=1}^{T} \widehat{\ell}_t\right] + v \cdot \mathbb{E}\left[\sum_{t=1}^{T} \ell_t\right],$$

527  where the first equality is due to the fact that $\widehat{\ell}_t$ is an unbiased estimator of $\ell_t$. We bound the last
528  term using the expression for $v$:

$$v \cdot \sum_{t=1}^{T} \ell_t = \sum_{t=1}^{T}\left[\sum_{i \ne i^\star} \gamma \ell_{t,i} - (N-1)\gamma \ell_{t,i^\star}\right] \le \gamma TN,$$

529  where in the last inequality we use the fact that the losses are bounded in $[0, 1]$.  ∎

530  We will also make use of general FTRL regret bound given by Theorem 3 (which we prove in
531  Appendix C) together with the stability and penalty bounds shown in the previous sections. Theorem 2
532  is restated here in the precise form proved below.

533  **Theorem 2** (restated)**.** *Algorithm 1 attains the following regret bound, regardless of the corruption
534  level, for $NT \ge 3^{11}$:*

$$\mathcal{R}_T \le 9K \log(NT) + 6\log^2(NT) \sum_{t=1}^{T} \sum_{k \ne k^\star} \sqrt{\frac{\mathbb{E}[p_t(V_k)]}{t}}$$

$$+ 2\log(NT) \sum_{t=1}^{T} \sqrt{\frac{\mathbb{E}[p_t(V_{k^\star} \setminus i^\star)]}{t}} + 16 \sum_{t=1}^{T} \sqrt{\frac{\mathbb{E}[p_t^+(V_{k^\star} \setminus i^\star)]}{t}}. \tag{13}$$

535  *Proof.* Note that due to Lemma 9 it suffices to bound $\mathbb{E}\left[\sum_{t=1}^{T}(p_t - p^\gamma) \cdot \widehat{\ell}_t\right]$ where $p^\gamma$ is defined by

$$p_i^\gamma = \begin{cases} \gamma & i \ne i^\star \\ 1 - (N-1)\gamma & i = i^\star \end{cases} \quad \forall i \in [N],$$

536  since it can only be larger than the pseudo-regret by an additive constant. Using Theorem 3 and then
537  bounding the penalty and stability terms using Lemma 4 and Lemma 5 we obtain

$$\mathcal{R}_T \le 9K \log(NT) + \left(5\log^2(NT) + 112\right) \sum_{t=1}^{T} \sum_{k \ne k^\star} \sqrt{\frac{\mathbb{E}[p_t(V_k)]}{t}}$$

$$+ 2\log(NT) \sum_{t=1}^{T} \sqrt{\frac{\mathbb{E}[p_t(V_{k^\star} \setminus i^\star)]}{t}} + 16 \sum_{t=1}^{T} \sqrt{\frac{\mathbb{E}[p_t^+(V_{k^\star} \setminus i^\star)]}{t}}$$

$$\leq 9K\log(NT) + 6\log^2(NT) \sum_{t=1}^{T} \sum_{k \neq k^\star} \sqrt{\frac{\mathbb{E}[p_t(V_k)]}{t}}$$

$$+ 2\log(NT) \sum_{t=1}^{T} \sqrt{\frac{\mathbb{E}[p_t(V_{k^\star} \setminus i^\star)]}{t}} + 16 \sum_{t=1}^{T} \sqrt{\frac{\mathbb{E}[p_t^+(V_{k^\star} \setminus i^\star)]}{t}},$$

where the last inequality holds for $NT \geq 3^{11}$. ∎

### B.5  Proof of Theorem 1 (Main)

We can now provide a proof of our main result given in Theorem 1, restated here more precisely.

**Theorem 1** (restated)**.** *Algorithm 1 attains the following expected pseudo-regret bound in the C-corrupted stochastic setting, for $NT \geq 3^{11}$:*

$$\mathcal{R}_T \leq 184\log^2(NT) \cdot \min\left\{ \sqrt{KT}, \log^2(NT) \sum_{k:\Delta_k>0} \frac{\log T}{\Delta_k} + \sqrt{C \sum_{k:\Delta_k>0} \frac{\log T}{\Delta_k}} \right\}.$$

*Proof.* We first prove the following:

$$\mathcal{R}_T \leq 184\log^4(NT) \sum_{k:\Delta_k>0} \frac{\log T}{\Delta_k} + 28\log^2(NT) \sqrt{C \sum_{k:\Delta_k>0} \frac{\log T}{\Delta_k}}.$$

We proceed bounding the RHS of Eq. (13). For all $B, z > 0$ we have

$$B \sum_{t=1}^{T} \left( \sum_{k \neq k^\star} \sqrt{\frac{\mathbb{E}[p_t(V_k)]}{t}} + \sqrt{\frac{\mathbb{E}[p_t(V_{k^\star} \setminus i^\star)]}{t}} \right) \leq B^2 \cdot z \sum_{t=1}^{T} \sum_{k:\Delta_k>0} \frac{1}{2t\Delta_k} + \frac{1}{2z} \sum_{t=1}^{T} \sum_{i=1}^{N} \mathbb{E}[p_{t,i}]\delta_i$$

$$\leq B^2 \cdot z \sum_{k:\Delta_k>0} \frac{\log T}{\Delta_k} + \frac{1}{2z} \sum_{t=1}^{T} \sum_{i=1}^{N} \mathbb{E}[p_{t,i}]\delta_i$$

$$\leq B^2 \cdot z \sum_{k:\Delta_k>0} \frac{\log T}{\Delta_k} + \frac{1}{2z}(\mathcal{R}_T + 2C), \qquad (14)$$

where the first inequality is due to Young's inequality and the fact that $\Delta_k \leq \delta_i$ for all $i \in V_k$, the second inequality is since $\sum_{t=1}^{T}(1/t) \leq 2\log T$ and the last inequality is due to the following simple observation which follows from the definition of corruption:

$$\mathbb{E}\left[ \sum_{t=1}^{T} \sum_{i=1}^{N} p_{t,i}(\tilde{\ell}_{t,i} - \tilde{\ell}_{t,i^\star}) \right] \leq \mathbb{E}\left[ \sum_{t=1}^{T} \sum_{i=1}^{N} p_{t,i}(\ell_{t,i} - \ell_{t,i^\star}) \right] + 2\mathbb{E}\left[ \sum_{t=1}^{T} \|\ell_t - \tilde{\ell}_t\|_\infty \right]$$

$$= \mathbb{E}\left[ \sum_{t=1}^{T} \sum_{i=1}^{N} p_{t,i}(\ell_{t,i} - \ell_{t,i^\star}) \right] + 2C.$$

Setting $B = 6\log^2(NT)$ gives a bound on the second term in the RHS of Eq. (13). Similarly, we have

$$16 \sum_{t=1}^{T} \sqrt{\frac{1}{t}\mathbb{E}[p_t^+(V_{k^\star} \setminus i^\star)]} \leq 256z \sum_{k:\Delta_k>0} \frac{\log T}{\Delta_k} + \frac{1}{2z} \sum_{t=1}^{T} \sum_{i=1}^{N} \mathbb{E}[p_{t,i}^+]\delta_i$$

$$\leq 256z \sum_{k:\Delta_k>0} \frac{\log T}{\Delta_k} + \frac{C}{z} + \frac{1}{2z}\mathbb{E}\left[ \sum_{t=1}^{T} \sum_{i=1}^{N} p_{t,i}^+ \cdot (\ell_{t,i} - \ell_{t,i^\star}) \right]. \qquad (15)$$

We now use Lemma 8 to bound the rightmost term of Eq. (15) as follows:

$$\mathbb{E}\left[\sum_{t=1}^{T}\sum_{i=1}^{N} p_{t,i}^{+} \cdot \left(\ell_{t,i} - \ell_{t,i^{\star}}\right)\right] = \mathbb{E}\left[\sum_{t=1}^{T} p_{t}^{+} \cdot \left(\mathbb{E}_t[\widehat{\ell}_t] - \ell_{t,i^{\star}}\mathbf{1}\right)\right]$$

$$\leq \mathbb{E}\left[\sum_{t=1}^{T} p_{t} \cdot \left(\mathbb{E}_t[\widehat{\ell}_t] - \ell_{t,i^{\star}}\mathbf{1}\right)\right]$$

$$= \mathbb{E}\left[\sum_{t=1}^{T}\sum_{i=1}^{N} p_{t,i} \cdot \left(\ell_{t,i} - \ell_{t,i^{\star}}\right)\right]$$

$$= \mathcal{R}_T,$$

where we used the fact that $\widehat{\ell}_t$ is an unbiased estimator for $\ell_t$. We can conclude that

$$16\sum_{t=1}^{T}\sum_{k:\Delta_k>0}\sqrt{\frac{1}{t}\mathbb{E}[p_t^+(V_k)]} \leq 256\sum_{k:\Delta_k>0}\frac{\log T}{\Delta_k} + \frac{1}{2z}(\mathcal{R}_T + 2C). \tag{16}$$

Using Theorem 2 and combining the bounds from Eq. (14) and Eq. (16) we obtain

$$\mathcal{R}_T \leq 9K\log(NT) + \left(36\log^4(NT) + 256\right)z\sum_{k:\Delta_k>0}\frac{\log T}{\Delta_k} + \frac{1}{z}\mathcal{R}_T + \frac{2C}{z}$$

$$\leq \left(45\log^4(NT) + 256\right)z\sum_{k:\Delta_k>0}\frac{\log T}{\Delta_k} + \frac{1}{z}\mathcal{R}_T + \frac{2C}{z}$$

$$\leq 46\log^4(NT)z\sum_{k:\Delta_k>0}\frac{\log T}{\Delta_k} + \frac{1}{z}\mathcal{R}_T + \frac{2C}{z},$$

where the second inequality is since $K \leq 1 + \sum_{k:\Delta_k>0} 1/\Delta_k$ and the last inequality holds since $NT \geq 3^4$. Rearranging and simplifying we obtain

$$\mathcal{R}_T \leq 2U + (z-1)U + \frac{2C+U}{z-1},$$

where we denote $U = 46\log^4(NT)\sum_{k:\Delta_k>0}\log T/\Delta_k$ for simplicity. We now choose $z$ which minimizes the bound, by setting $z = 1 + \sqrt{\frac{U+2C}{U}}$. This gives us

$$\mathcal{R}_T \leq 2U + 2\sqrt{U(U+2C)}$$

$$\leq 4U + 4\sqrt{UC}$$

$$\leq 184\log^2(NT)\sum_{k:\Delta_k>0}\frac{\log T}{\Delta_k} + 28\log^2(NT)\sqrt{C\sum_{k:\Delta_k>0}\frac{\log T}{\Delta_k}},$$

which concludes the first part of the proof. We now show that

$$\mathcal{R}_T \leq 28\log^2(NT)\sqrt{KT}.$$

We again use Theorem 2 and also the fact that $p_t^+(V_k) \leq \frac{7}{3}p_t(V_k)$ by Lemma 7, to obtain

$$\mathcal{R}_T \leq 9K\log(NT) + \left(6\log^2(NT) + 32\right)\sum_{t=1}^{T}\frac{1}{\sqrt{t}}\sum_{k=1}^{K}\sqrt{p_t(V_k)}$$

$$\leq 9K\log(NT) + 7\log^2(NT)\sum_{t=1}^{T}\frac{1}{\sqrt{t}}\sum_{k=1}^{K}\sqrt{p_t(V_k)},$$

where the inequality holds since $NT \geq 3^6$. We conclude the proof via the following straightforward calculation:

$$\sum_{t=1}^{T}\frac{1}{\sqrt{t}}\sum_{k=1}^{K}\sqrt{p_t(V_k)} \leq \sqrt{K}\sum_{t=1}^{T}\frac{1}{\sqrt{t}} \leq 2\sqrt{KT},$$

where we used Jensen's inequality and the fact that $\sum_{t=1}^{T}(1/\sqrt{t}) \leq 2\sqrt{T}$. We obtained two regret bounds and thus the minimum of the two holds, which concludes the proof. ∎

## C Refined Regret Bound for FTRL

Consider the FTRL framework which generates predictions $w_1, w_2, ..., w_T \in \mathcal{W}$ given a sequence of arbitrary loss vectors $g_1, g_2, ..., g_T$ and a sequence of regularization functions $H_1, H_2, ..., H_T$. The following gives a general regret bound which we use in order to prove Theorem 2.

**Theorem 3.** *Suppose $H_t = \eta_t^{-1}\psi + \phi$ for twice-differentiable and convex functions $\psi$ and $\phi$, $\psi$ being strictly convex. Let $w_t^+ = \arg\min_{w \in \mathcal{W}}\{w \cdot \sum_{s=1}^t g_s + H_t(w)\}$. Then there exists a sequence of points $\tilde{w}_t \in [w_t, w_t^+]$ such that, for all $w^* \in \mathcal{W}$:*

$$\sum_{t=1}^T g_t \cdot (w_t - w^\star) \le \phi(w^\star) - \phi(w_1) + \sum_{t=1}^T \left(\frac{1}{\eta_t} - \frac{1}{\eta_{t-1}}\right)(\psi(w^\star) - \psi(w_t)) + 2\sum_{t=1}^T \eta_t \left(\|g_t\|_t^*\right)^2.$$

*Here $\|g\|_t = \sqrt{g^\mathsf{T}\nabla^2\psi(\tilde{w}_t)g}$ is the local norm induced by $\psi$ at $\tilde{w}_t$, and $\|\cdot\|_t^*$ is its dual. Here we also define $1/\eta_0 \triangleq 0$.*

*Proof.* We directly follow an analysis by Jin and Luo [13], and include the details for completeness. For simplicity we denote $G_t = \sum_{s=1}^t g_s$. We make the following definitions:

$$F_t(w) = w \cdot G_{t-1} + H_t(w),$$
$$F_t^+(w) = w \cdot G_t + H_t(w),$$

such that $w_t = \arg\min_{w \in \mathcal{W}}\{F_t(w)\}$ and $w_t^+ = \arg\min_{w \in \mathcal{W}}\{F_t^+(w)\}$. Fix $w^\star \in \mathcal{W}$. We note that the regret of FTRL with respect to $w^\star$ has the following decomposition:

$$\sum_{t=1}^T g_t \cdot (w_t - w^\star) = \sum_{t=1}^T (w_t \cdot g_t + F_t(w_t) - F_t^+(w_t^+)) + \sum_{t=1}^T (F_t^+(w_t^+) - F_t(w_t) - w^\star \cdot g_t).$$

We first show that for all time steps $t$ it holds that

$$w_t \cdot g_t + F_t(w_t) - F_t^+(w_t^+) \le 2\eta_t \left(\|g_t\|_t^*\right)^2. \tag{17}$$

We lower bound $w_t \cdot g_t + F_t(w_t) - F_t^+(w_t^+)$ as follows:

$$\begin{aligned} w_t \cdot g_t + F_t(w_t) - F_t^+(w_t^+) &= w_t \cdot G_t + H_t(w_t) - F_t^+(w_t^+) \\ &= F_t^+(w_t) - F_t^+(w_t^+) \\ &= \nabla F_t^+(w_t^+) \cdot (w_t - w_t^+) + \frac{1}{2}\|w_t - w_t^+\|_{\nabla^2 H_t(\tilde{w}_t)}^2 \\ &\ge \frac{1}{2}\|w_t - w_t^+\|_{\nabla^2 H_t(\tilde{w}_t)}^2 \\ &\ge \frac{1}{2}\eta_t^{-1}\|w_t - w_t^+\|_t^2, \end{aligned}$$

where the third line is a Taylor expansion of $F_t^+$ around $w_t^+$, with $\tilde{w}_t$ being a point between $w_t$ and $w_t^+$, in the second to last line we use a first-order optimality condition of $w_t^+$, and in the last line we use the fact that $\nabla^2 H_t \ge \eta_t^{-1}\nabla^2\psi$. We now upper bound $w_t \cdot g_t + F_t(w_t) - F_t^+(w_t^+)$ as follows:

$$\begin{aligned} w_t \cdot g_t + F_t(w_t) - F_t^+(w_t^+) &= (w_t - w_t^+) \cdot g_t + F_t(w_t) - F_t(w_t^+) \\ &\le (w_t - w_t^+) \cdot g_t \\ &\le \left(\sqrt{\eta_t^{-1}}\|w_t - w_t^+\|_t\right)\left(\sqrt{\eta_t}\|g_t\|_t^*\right) \\ &= \|w_t - w_t^+\|_t \cdot \|g_t\|_t^*, \end{aligned}$$

where in the first inequality we use the fact that $w_t$ is the minimizer of $F_t$ and the second inequality is an application of Hölder's inequality. Combining the lower and upper bounds gives us Eq. (17). Next we show that

$$\sum_{t=1}^T (F_t^+(w_t^+) - F_t(w_t) - w^\star \cdot g_t) \le \phi(w^\star) - \phi(w_1) + \sum_{t=1}^T \left(\frac{1}{\eta_t} - \frac{1}{\eta_{t-1}}\right)(\psi(w^\star) - \psi(w_t)). \tag{18}$$

We bound the LHS of Eq. (18) as follows:

$$\sum_{t=1}^{T} \left( F_t^+(w_t^+) - F_t(w_t) - w^\star \cdot g_t \right)$$

$$\leq -F_1(w_1) + \sum_{t=2}^{T} \left( F_{t-1}^+(w_t) - F_t(w_t) \right) + F_T^+(w_T^+) - w^\star \cdot G_T$$

$$\leq -F_1(w_1) + \sum_{t=2}^{T} \left( F_{t-1}^+(w_t) - F_t(w_t) \right) + F_T^+(w^\star) - w^\star \cdot G_T$$

$$= -H_1(w_1) - \sum_{t=2}^{T} \left( \frac{1}{\eta_t} - \frac{1}{\eta_{t-1}} \right) \psi(w_t) + H_T(w^\star)$$

$$= -\eta_1^{-1} \psi(w_1) - \phi(w_1) - \sum_{t=2}^{T} \left( \frac{1}{\eta_t} - \frac{1}{\eta_{t-1}} \right) \psi(w_t) + \eta_T^{-1} \psi(w^\star) + \phi(w^\star)$$

$$= \phi(w^\star) - \phi(w_1) + \sum_{t=1}^{T} \left( \frac{1}{\eta_t} - \frac{1}{\eta_{t-1}} \right) \left( \psi(w^\star) - \psi(w_t) \right),$$

where in the first and second inequalities we use the optimality of $w_t^+$. Combining Eq. (17) and Eq. (18) we conclude the proof. ∎

*Proof of Lemma 3.* Fix any $p^\gamma \in \mathcal{S}_N^\gamma$. The lemma follows immediately by applying Theorem 3 to Algorithm 1 with the regularizations $R_1, R_2, ..., R_T$ and the shifted loss estimators $\hat{\ell}_t - \ell_{t,i^\star} \mathbf{1}$, while noting that constant shifts in the loss estimators do not change the algorithm whatsoever. ∎