# OpenReview forum: "Towards Best-of-All-Worlds Online Learning with Feedback Graphs"
_NeurIPS.cc/2021/Conference — NeurIPS 2021 Poster_

### Official Review · Reviewer_bWJC · 2021-07-16

**Rating:** 6
**Confidence:** 3

**Summary:**

This paper considers the problem of online learning with feedback graphs. Feedback graphs tell which losses the learner gets to observe each time an action is chosen.  In particular, special cases include the bandit setting when the graph has no edges, and the full information setting when the feedback graph is a clique.

The authors derive bounds that scale in terms of the clique covering number of the graph. This clique covering number is the number of cliques (fully connected subgraphs) that are appearing in the graph. This number is useful because each time one arm is picked, the learner can observe the losses of all the other arms that are in the same clique.

Reasoning about the problem in terms of cliques is interesting because the exploration vs exploitation trade-off is not the same everywhere on the graph. If we consider each clique as one entity, we have a standard multi-armed bandits exploration vs exploitation trade-off. But then, when considering which action to play within the clique, this is a full information problem as we will get the feedback for all actions within the clique either way.

The authors then provide a FTRL algorithm that combines the Tsallis entropy (optimal for MAB) and the Shannon entropy (optimal for Full info).  They also mix in a log barrier regularizer to help with the stability of the algorithm.

The authors derive a single bound that covers simultaneously the adversarial, the stochastic, and the corrupted setting.
That bound is near-optimal because of the consideration of the covering clique number rather than the independence number (which basically means that the edges in the feedback graph that are not part of a clique are not taken advantage of), and because the extra log barrier regularization term leads to extra log factors.

**Limitations And Societal Impact:**

There is no conclusion and the limitations of the work are not clearly discussed. In particular, what is the intuition regarding closing the gap between the bound scaling in terms of the covering clique number, and the lower bound scaling with the independence number.

Other limitations that should be discussed:
The uniqueness of the best arm in intermediate settings.
Not considering the stochastically constrained adversarial regime. In the MAB setting, Tsallis-Inf is capable to achieve the stochastic bounds in this more general setting. Do the authors think that something similar could be done here? A best-of-all-worlds setting should probably consider that regime too.


**Main Review:**

The problem studied is a combination of the Best-of-All-Worlds line of work that is been widely studied in the past couple of years in multi-armed bandits with the bandits with feedback graphs line of work. This combination appears to be novel.
To be completely best of all worlds, the authors may have considered the stochastically constrained adversarial regime rather than solely the stochastic regime.

The algorithm derived builds upon the widely used FTRL framework and combines different already existing regularizers. Those regularizers have proved to be optimal in their respective settings (Tsallis for the MAB setting and Shannon for the Full info), and they are combined in a way that exploits the feedback graph structure efficiently.
The results achieved are interesting, even though they are not actually optimal.


The paper is clearly written and it is nice that some proof sketches are given in the body of the paper.
This being said some final discussion about the work is lacking, in particular about the limitations of the work (see the next section for details).

**Time Spent Reviewing:**

8

---

> ### Author Response · Authors · 2021-08-10
> **Thanks for the review - author response**
>
> Many thanks for the thorough review. We address the main points and concerns below:
>
> > **“the extra log barrier regularization term leads to extra log factors”**
>
> Note that the extra log factors in fact do not arise from the log-barrier term but rather from our main regularizer: to ensure its strong convexity, we had to set $\alpha$ of the form $\log^2(NT)$ in Eq. (2) (see statement of Lemma 2), which leads to an $O(\log^4(T))$ factor in the final bound.
>
>
> > **“limitations that should be discussed”**
>
> Thank you for pointing those out---we will include a discussion in the final version, along the lines of our responses below.
>
>
> > **“the authors may have considered the stochastically constrained adversarial regime”**
>
> We did not explicitly consider the adversarially constrained stochastic regime, but we believe that our results extend in a straightforward manner to that setting (as is the case in MAB).  Note that we did consider a related setting that received much more significant focus recently: the adversarially-corrupted stochastic setting, where we obtain the first non-trivial results for graph-based feedback.
>
>
> > **“what is the intuition regarding closing the gap between the bound scaling in terms of the covering clique number, and the lower bound scaling with the independence number”**
>
> Our belief is that this would be possible with a similar FTRL-based approach, with a similar bi-level regularization which will use the neighborhood probabilities of the nodes instead of the clique marginal probabilities (we will definitely mention this explicitly as an open problem in the final version and detail some of the basic intuitions we have).  However, such a regularization lacks the hierarchical structure required by our current analysis of the Hessian and appears to require further ideas; in particular, we do not know what is the analogous version of Lemma 1 (which is used to establish the crucial strong convexity property) for this generalized regularization.  We do believe that our paper makes significant progress in terms of techniques towards establishing the optimal bounds in this setting.
>
>
> > **“The uniqueness of the best arm in intermediate settings”**
>
> Uniqueness of the best arm is already crucial in best-of-both-worlds analyses in the basic MAB case (e.g., Zimmert & Seldin ‘19 [27]).  Since our analysis of the intermediate setting (with corruption budget $C$) builds on the same ideas used in the purely stochastic setting, this assumption is also crucial there. It is an interesting question whether this assumption can be avoided---already in the basic MAB case---we believe that it is merely an artifact of the proof technique that relies on self-bounding the regret.

---

### Official Review · Reviewer_ZV4A · 2021-07-16

**Rating:** 6
**Confidence:** 4

**Summary:**

This paper studies a best-of-all-worlds algorithm for online learning with feedback graphs by incorporating a novel regularizer. The proposed algorithm simultaneously achieves near-optimal regret w.r.t. $T$ in the adversarial setting, stochastic setting and corrupted stochastic setting.

**Limitations And Societal Impact:**

See main review.

**Main Review:**

It is an interesting problem to find a best-of-all worlds algorithm for online learning with feedback graphs. This paper presents some solid results, but I still have some concerns:

1. The algorithm requires to know the minimum clique covering of the graph, which is impractical in many applications. Also, most previous works on online learning with feedback graphs do not have this requirement.

2. As discussed in section 1.1, the current regret bounds are not optimal.

3. This paper only studies the case where the graph is strongly observable. Is it possible to generalize the results to the case where the graph is weakly observable? (see [1] for the definition of strongly observable and weakly observable)

4. Though the paper mainly focuses on the theoretical results, I suggest the authors to show some numerical results as previous best-of-all-worlds papers [13,27].

**Time Spent Reviewing:**

5

---

> ### Author Response · Authors · 2021-08-10
> **Thanks for the review - author response**
>
> Many thanks for the feedback---we address your main points below.
>
> > **“The algorithm requires to know the minimum clique covering of the graph”;  “the current regret bounds are not optimal”;  “This paper only studies the case where the graph is strongly observable”**
>
> Agreed - these are all indeed limitations of our algorithm, which we currently do not know how to avoid with our approach.  However, we believe that the results we were able to obtain already constitute a very significant step towards achieving optimal best-of-all-worlds results in the feedback graph setting, and already required a fair amount of sophistication to obtain.  In particular, we believe that our novel bi-level regularization and the intricate analysis of its Hessian (detailed in Section 3) would prove as key techniques towards a full resolution of the best-of-all-worlds problem.   The points you raised are all very interesting questions for future research.
>
> See also our reply to Reviewer bWJC for more details regarding our intuition about closing the gap between the upper bounds scaling with the clique covering number and the lower bound scaling with the independence number.

---

> > ### Comment · Reviewer_ZV4A · 2021-09-10
> > **Reply**
> >
> > Thanks to the authors for the reply. After reading the author response and other reviews, I agree that this paper makes some important progress to a difficult problem. I increased my score to 6.

---

### Official Review · Reviewer_jtgp · 2021-07-17

**Rating:** 7
**Confidence:** 3

**Summary:**

This paper introduces a novel bi-level regularization, namely the Tsallis-Shannon entropy, which is a combination of the two types of entropy with a linear shift. The paper proves the lower bound for the Hessian of the new regularization which is a diagonal matrix with positive entries, and thus convexity. By applying the regularization in the "Follow the Regularized Leader" framework, the paper derives an algorithm for online learning problems with undirected graph feedback, which is a generalization of the Multi-Armed Bandits (MAB) problem and Prediction with Expert Advice (PEA) problem. The algorithm achieves near optimal regret bounds for both bandits and full feedback problems with adversarially-corrupted stochastic loss setting. The paper also proves a bound for online learning problems with general undirected graph feedback, which is a big step towards a best-of-all-worlds guarantee for this setting.

**Limitations And Societal Impact:**

Yes.

**Main Review:**

Strengths:
1. The regularization proposed in the paper is new and interesting. It seems that this is the first multi-level regularization in such a form that provably achieves best-of-all-worlds type guarantees in online learning.
2. The algorithm proposed achieves highly non-trivial regret bounds.
3. The proofs seem quite non-trivial and the ideas might be useful for solving other similar problems.
4. The writing is clear. It is very helpful that high level ideas and motivations are provided for the technical proofs, making them easier to understand.



Weaknesses:
1. The algorithm requires a clique cover of the feedback graph as input but finding a minimum clique cover is hard. In Alon et al. [1], the algorithm does not need such information as input, and the independence number in the regret bounds is not required to be known in advance.
2. The bound in this paper depends on the clique covering number of the feedback graph, which is weaker than the independence number of the graph in Alon et al. [1]. So it might be sub-optimal even in the stochastic setting.
3. Some minor typos, e.g. line 95, line 240.

Question:
I am confused about the derivation between line 501 and 502 in the appendix (the 3rd and 4th inequality). Can you provide more details?


**Time Spent Reviewing:**

4

---

> ### Author Response · Authors · 2021-08-10
> **Thanks for the review - author response**
>
> Thank you for the knowledgeable review and insightful comments!  We first address the main concerns; specific replies follow below.
>
> > **“The algorithm requires a clique cover of the feedback graph as input but finding a minimum clique cover is hard”; “The bound in this paper depends on the clique covering number of the feedback graph, which is weaker than the independence number”**
>
> We agree that these are limitations of our algorithm, and ones we would have avoided had we known how to (we tried to be completely upfront about these limitations in our exposition - do let us know if you found any issues).  However, we also feel that the results we were able to obtain are already significant and far from being trivial (as you seem to agree!).  It is a very interesting question whether similar best-of-all-worlds bounds are possible without non-trivial pre-processing of the graph, and an even more interesting question whether one could obtain optimal best-of-all-worlds bounds that scale with the independence number.
>
>
> **As for the technical question:** the derivation between line 501 and 502 (specifically the 3rd and 4th inequality) follows from the fact that for any set of vertices $A$ it holds that $\tilde{p}_t(A) \leq p_t(A) + p_t^+(A)$, as $\tilde{p}_t$ lies between $p_t$ and $p_t^+$ (coordinate-wise), and from the fact that $\sqrt{a+b} \leq \sqrt{a} + \sqrt{b}$ for $a,b \geq 0$.  We will be happy of course to clarify further if needed during the discussion.

---

> > ### Comment · Reviewer_jtgp · 2021-08-28
> > **Response to rebuttal**
> >
> > Dear authors,
> >
> > Thank for the clarification.
> >
> > During the discussion, I were aware that the theoretical results is actually weaker than I thought. But I still think the paper provides interesting insights and make non-trivial contributions to a difficult and important problem. So I change my score from 8 to 7.

---

> > > ### Author Response · Authors · 2021-08-28
> > > **Thanks for the response**
> > >
> > > Dear reviewer - thank you for keeping us posted on your updated reassessment.
> > >
> > > We actually found your earlier summary of the paper’s strengths and weaknesses rather spot-on; we would greatly appreciate it if you could elaborate a bit more as to what in the discussion has made you now feel that the results are weaker than you initially thought, so as to give us a chance to respond properly before discussions are finalized.  Thanks!

---

### Official Review · Reviewer_3tpU · 2021-07-18

**Rating:** 6
**Confidence:** 4

**Summary:**

The authors consider the best-of-both-worlds problem in bandits with graph feedback. Similar to [27], they consider stochastic bandits with C adversarial corruption that interpolates between the stochastic setting and the adversarial setting . As a result, they get $\text{polylog}( T)\sum_{k\in K}1/\Delta_{\min,k}$ regret in the stochastic setting and $\sqrt{|K|T}$ in the adversarial setting, where K is a clique cover of G.

**Main Review:**

- The paper in general is well written, with detailed explanations on deriving the regularizer and concise but informative proof sketch. Although the self-bounding technique, the constant log-barrier term, and some lemmas all appeared in previous works already, combining these together and getting this novel result in bandits with graph-feedback is still highly non-trivial.

- My main concern is on the stochastic part. Basically we have $O(\log^5(T))$ dependence on the time horizon (if I count correctly), which doesn't look very optimal even if we consider the best-of-both-worlds setting. It looks like the authors hide this and use polylog most of the time, which is fine, but I may would like to hear from the authors regarding the tightness of this bound under the current algorithm and the analysis.

- Related to the comment above, I found the authors tend to represent stochastic bounds in terms of $O(\sum_{i\in S}\log(T)/\Delta_{i})$ for some set S as $O(|S|\log(T))$. This seems not a standard representation in the literature, and in fact, technically, not correct. The $1/\Delta_i$ term can be very large, which is independent of how many i you sum over. Therefore, it doesn't seem right to me the authors simply say the optimal bound is $O(\alpha (G) \log T)$ while theirs is $O(\theta (G) \text{polylog}~T)$. A more careful comparison between the bound in this paper and the previous work would be helpful for the reviewers to understand the significance. Some papers on stochastic graph bandits such as [1] may also be a paper that the authors can compare with.

- Lemma 1 looks very technical and the authors don't prove it in the main text, either. Thus, It is not very informative to me and I would suggest move it somewhere else and show 3.2 first.

- [17] also uses the clique partition in their Algorithm 1 and the constant log-barrier term. I think the authors may also compare this with their approach.  Some natural question is like whether their algorithm can achieve best-of-both-worlds when replacing the log-barrier with Tsallis entropy. (Note that the corralling trick is also possible using the method in [2])

- As K is commonly used as the number of arms in MAB, I think using $K=\theta(G)$ is a little bit confusing. Using $\theta$ as a shorthand of $\theta(G)$ is clearer to me. $\kappa$ is also another commonly used notation for this quantity.

[1] Lykouris, Thodoris, Éva Tardos, and Drishti Wali. "Feedback graph regret bounds for Thompson Sampling and UCB."

[2] Foster, Dylan J., Claudio Gentile, Mehryar Mohri, and Julian Zimmert. "Adapting to misspecification in contextual bandits."

**Time Spent Reviewing:**

5

---

> ### Author Response · Authors · 2021-08-10
> **Thanks for the review - author response**
>
> Thank you for the thorough and informative review!  We hope that our replies below will help in clarifying your main concerns.
>
> > **“we have $O(\log^5⁡(T))$ dependence on the time horizon (if I count correctly), which doesn't look very optimal”**
>
> Indeed there is an $O(\log^5(T))$ dependence on the time horizon in the stochastic setting, which is a $\log^4(T)$ factor away from the optimal $O(\log(T))$. Our intention was not to hide this, but to make the bounds more succinct by writing $\text{polylog}(T)$. However, this is a valid concern and this dependence is indeed not optimal, especially in the purely stochastic setting---we will be sure to make this more explicit in the revision.
>
> We believe that our algorithm as currently implemented achieves these bounds tightly. In a bit more detail, the $\log^4(T)$ extra factor arises from multiplying the standard Tsallis entropy over the clique marginals by a term scaling like $\log^2(T)$ and then the self-bounding argument makes it grow to $\log^4(T)$. While we do believe that this dependence may be improved with some additional tricks and technical improvements to the algorithm, our focus was on nailing down the best possible dependence on the graph parameters while still obtaining (poly)logarithmic scaling w.r.t. $T$.
>
>
> > **“the authors tend to represent stochastic bounds in terms of $O(\sum_{i \in S} \log(T) / \Delta_i)$ for some set $S$ as $O(|S| \log⁡(T))$. This seems not a standard representation in the literature, and in fact, technically, not correct”**
>
> Indeed, this is a somewhat informal way of writing regret bounds, and one that we used only in our introduction---the technical sections state the actual, formal bounds with the precise dependence on the gaps $\Delta_i$. In light of your comment, we will reconsider this presentation and think of a better alternative for the final version.  Our motivation for writing the bounds in such a form is to give a clear sense of how the regret scales in terms of the graph properties, and to contrast it with analogous bounds that roughly scale with the independence number.  Indeed, the optimal rate as established in prior work is of the form $O(\sum_{i \in S} \log(T) / \Delta_i)$ where $S$ is a set of at most $O(\alpha(G))$ arms, and bounds of this form can also be seen in [1]. (We neglected to cite the latter paper in the submission---will be fixed in the final version, thanks!)
>
>
> > **“Lemma 1 is not very informative to me and I would suggest move it somewhere else and show 3.2 first”**
>
> You are correct that Lemma 1 is somewhat uninformative before seeing Lemma 2, being the main result of Section 3 that establishes the strong convexity property essential to our analysis. We will revise and start with the statement of Lemma 2. Thanks for this suggestion.
>
>
> > **“[17] also uses the clique partition in their Algorithm 1 and the constant log-barrier term. I think the authors may also compare this with their approach”**
>
> Indeed, we should have discussed the relation to this paper in some more detail---we will do so in the revision.  Note that the log-barrier [17] uses is over individual arm probabilities, whereas we crucially use a variant of the log-barrier taken over the marginal clique probabilities.  This is in order to avoid (additive) dependence of $O(N)$ in the regret, which is suboptimal in the feedback graph setting ([17] indeed suffers such an additive term).
>
> > **“Some natural question is whether their algorithm can achieve best-of-both-worlds when replacing the log-barrier with Tsallis entropy”**
>
>
> Good question: replacing this term with a Tsallis entropy would introduce an additive term scaling like $\sqrt{N}$ in the regret (specifically in the penalty term), which is suboptimal for the graph feedback setting and which we tried to avoid.
>
>
> > **“$K$ is commonly used as the number of arms in MAB”**
>
> Agreed - we will revise this notation. Thanks!

---

> > ### Comment · Reviewer_3tpU · 2021-08-18
> > **Thanks for the response**
> >
> > I thank the authors for the responses. It is good to know the authors plan to revise the paper based on my comments. I wanted to add that my main concern on the $O(|S|\log(T))$ expression was not only because this is informal, but it hides the dependence on the arms. When the bound is $O(\alpha (G) \log T)$, it may be fine as we can think of the summation is over the arms in an independent set. However, when the bound is $O(\theta (G) \log T)$, I cannot get the right dependence immediately from this expression as each clique is a "set". There are many possibilities such as the minimum, maximum, or average gap of the clique. It turns out the dependence of this paper is the worst one, that is, the minimum gap. I guess this is not a big issue, but I think that is another reason why the authors should be more careful when using the $O(\theta (G) \log T)$ notation.
> >
> > > Good question: replacing this term with a Tsallis entropy would introduce an additive term scaling like $\sqrt{N}$ in the regret (specifically in the penalty term), which is suboptimal for the graph feedback setting and which we tried to avoid.
> >
> > I am not sure I understand this. Let me be specific. I was wondering if we can replace the first summation of Eq. (7) in [17] with a Tsallis entropy. As the summation is over clique covers, the additive term should scale like $\sqrt{\theta(G)}$ instead of $\sqrt{N}$ ? Anyway, I think the answer to this question may also be something good to add to the paper.

---

> > > ### Author Response · Authors · 2021-08-19
> > > **Thanks for the updated comments**
> > >
> > > Thank you very much for your response. Your comments are invaluable for improving our exposition and discussion, and we will definitely incorporate them in the final version.
> > >
> > > **Regarding the $O(\theta(G) \log T)$ bounds:**
> > >
> > > We feel we now better understand your point from your initial review. However, we would argue that both $O(\theta(G) \log T)$ and $O(\alpha(G) \log T)$ bounds “hide” the dependence on the arms in a rather similar way.  As you correctly recognized, the $\theta(G)$ actually stands for the *worst-case* summation over a choice of representatives from each clique in a minimum clique covering (i.e., those arms having the minimal gap within their clique). However, the $O(\alpha(G) \log T)$-type bounds also involve a worst-case summation: the $\alpha(G)$ appearing in the bound is in fact the *maximum over all independence sets* of a summation over arms in the independence set.  To illustrate the point, if we consider the case where the graph is a union of disjoint cliques (when $\alpha(G)$ and $\theta(G)$ are equal) then a worst case choice of a maximum independent set *is equivalent to* a worst case choice of clique representatives, and so the two bounds exactly coincide.
> > >
> > > Having said all of the above, we do agree that this is quite subtle and we will make sure to clearly remark and discuss this point when presenting our bounds. Thanks again for pointing out!
> > >
> > > **Replacing log-barrier in Eq. (7) [17] with a Tsallis entropy:**
> > >
> > > Indeed, we misunderstood your question from earlier---apologies for our confusion.
> > > The paper [17] assumes an hierarchical approach which maintains an explicit probability distribution over the cliques which is updated via OMD steps, whereas the individual arm probabilities within each clique are handled by a dedicated instance of adaptive Hedge.
> > > This is a natural approach for our problem as well, and one that we indeed initially attempted at (with a Tsallis entropy regularization on the marginal clique probabilities, along the lines you suggested).  However, as the authors in [17] remark themselves (this is also discussed in Agarwal et al. (2017) cited therein), this approach is considerably challenging to analyze in partial information settings.  We note that best-of-both-worlds bounds themselves are tricky to analyze (even in multi-armed bandits), so combining the two analysis techniques did not seem very promising to us at that point.  It might very well be the case that this kind of approach can indeed give best-of-both-worlds results, though we do now know how to analyze it.  In any case, this seems like something worth discussing in the final version---thank you once again!

---

> > > > ### Comment · Reviewer_3tpU · 2021-08-25
> > > > **Thanks for the response**
> > > >
> > > > I thank the authors again for the responses. I agree what the authors point out regarding the $O(\theta(G) \log T)$ bound. Yet another point is that we cannot choose an optimal clique cover as it depends on the unknown gaps even with infinite computation. Therefore, in worst case, we get "maximum over all clique covers of a summation over minimum-gap arms in the cliques". Anyway, I think I am fine with this bound if the authors remark and discuss these points properly.

---

### Decision · Program_Chairs · 2021-09-27

**Decision:**

Accept (Poster)

**Comment:**

This paper design a novel, bi-level regularization procedure to successfully achieve a best-of-all-worlds regret guarantee. I’ll start with the negatives and them move on to the positives.

On the negative side, all of the reviewers acknowledge that the results in the stochastic setting are suboptimal. Specifically, it is unclear if the exponent of 5 on the $\log(T)$ term is essential or if this dependence could be reduced (it does seem suboptimal). Another issue is that the authors’ algorithm is restricted to a fixed clique covering. All the reviewers agreed that this is a significant disadvantage, as the choice of clique covering influences the stochastic setting regret bound (by way of the minimum gaps in the corresponding cliques), and hence it is not possible for the algorithm to select the optimal clique covering (since the gaps are unknown). This is one reason why some assessments were lower than they originally were. In my view, this is the most significant weakness of this work. Also, the current regret bounds are based on clique coverings, whereas it would be desirable to obtain results based on independence number.

On the positive side, there are important technical advances in this work. All the reviewers (and I) thought that this is a very difficult problem, and the authors do make important first progress on this problem. Their contribution has significant novelty, and even if they have not yet found the right regularizer for this problem, they have taken an important step in this direction. So, to be clear, even making this progress for a fixed clique covering is considered to be significant. Therefore, this paper meets the bar for publication at NeurIPS, and it will be exciting to see what additional progress can be made going forward. For the final version, I would like to stress that the authors follow the advice of reviewer 3tpU regarding avoiding the informal presentation by way of $\theta(G)$. Not all clique coverings are the same, and among clique coverings of the same size, the corresponding regret bounds (based on the minimum gaps) could be very different. On that note, I encourage the authors to mention adaptivity to the best clique covering a good direction for future work.